# SurvITE: Learning Heterogeneous Treatment Effects from Time-to-Event Data

**Alicia Curth**[*]
University of Cambridge
amc253@cam.ac.uk

**Changhee Lee**[*]
Chung-Ang University
changheelee@cau.ac.kr

**Mihaela van der Schaar**
University of Cambridge
University of California, Los Angeles
The Alan Turing Institute
mv472@cam.ac.uk

## Abstract

We study the problem of inferring heterogeneous treatment effects from time-to-event data. While both the related problems of (i) estimating treatment effects for *binary or continuous* outcomes and (ii) *predicting survival* outcomes have been well studied in the recent machine learning literature, their combination – albeit of high practical relevance – has received considerably less attention. With the ultimate goal of reliably estimating the effects of treatments on instantaneous risk and survival probabilities, we focus on the problem of learning (discrete-time) treatment-specific conditional hazard functions. We find that unique challenges arise in this context due to a variety of covariate shift issues that go beyond a mere combination of well-studied confounding and censoring biases. We theoretically analyse their effects by adapting recent generalization bounds from domain adaptation and treatment effect estimation to our setting and discuss implications for model design. We use the resulting insights to propose a novel deep learning method for treatment-specific hazard estimation based on balancing representations. We investigate performance across a range of experimental settings and empirically confirm that our method outperforms baselines by addressing covariate shifts from various sources.

## 1 Introduction

The demand for methods evaluating the effect of treatments, policies and interventions on *individuals* is rising as interest moves from estimating population effects to understanding effect heterogeneity in fields ranging from economics to medicine. Motivated by this, the literature proposing machine learning (ML) methods for estimating the effects of treatments on continuous (or binary) end-points has grown rapidly, most prominently using tree-based methods [1, 2, 3, 4, 5], Gaussian processes [6, 7], and, in particular, neural networks (NNs) [8, 9, 10, 11, 12, 13, 14, 15]. In comparison, the ML literature on heterogeneous treatment effect (HTE) estimation with time-to-event outcomes is rather sparse. This is despite the immense practical relevance of this problem – e.g. many clinical studies consider time-to-event outcomes; this could be the time to onset or progression of disease, the time to occurrence of an adverse event such as a stroke or heart attack, or the time until death of a patient.

In part, the scarcity of HTE methods may be due to time-to-event outcomes being inherently more challenging to model, which is attributable to two factors [16]: (i) time-to-event outcomes differ from standard regression targets as the main objects of interest are usually not only expected survival times but the *dynamics of the underlying stochastic process*, captured by hazard and survival functions, and (ii) the presence of *censoring*. This has led to the development of a rich literature on survival analysis particularly in (bio)statistics, see e.g. [16, 17]. Classically, the effects of treatments in clinical studies with time-to-event outcomes are assessed by examining the coefficient of a treatment

---

[*]Equal contribution

35th Conference on Neural Information Processing Systems (NeurIPS 2021).

indicator in a (semi-)parametric model, e.g. Cox proportional hazards model [18], which relies on the often unrealistic assumption that models are correctly specified. Instead, we therefore adopt the nonparametric viewpoint of van der Laan and colleagues [19, 20, 21, 22] who have developed tools to incorporate ML methods into the estimation of treatment-specific *population average* parameters. Nonparametrically investigating treatment effect *heterogeneity*, however, has been studied in much less detail in the survival context. While a number of tree-based methods have been proposed recently [23, 24, 25, 26], NN-based methods lack extensions to the time-to-event setting despite their successful adoption for estimating the effects of treatments on other outcomes – the only exception being [27], who directly model event times under different treatments with generative models.

Instead of modeling event times directly like in [27], we consider adapting machine learning methods, with special focus on NNs, for estimation of (discrete-time) treatment-specific hazard functions. We do so because many target parameters of interest in studies with time-to-event outcomes are functions of the underlying temporal dynamics; that is, hazard functions can be used to directly compute (differences in) survival functions, (restricted) mean survival time, and hazard ratios. We begin by exploring and characterising the unique features of the survival treatment effect problem within the context of empirical risk minimization (ERM); to the best of our knowledge, such an investigation is lacking in previous work. In particular, we show that learning treatment-specific hazard functions is a challenging problem due to the potential presence of *multiple* sources of *covariate shift*: (i) non-randomized treatment assignment (confounding), (ii) informative censoring and (iii) a form of shift we term *event-induced* covariate shift, all of which can impact the quality of hazard function estimates. We then theoretically analyze the effects of said shifts on ERM, and use our insights to propose a new NN-based model for treatment effect estimation in the survival context.

**Contributions** (i) We identify and formalize key challenges of heterogeneous treatment effect estimation in time-to-event data within the framework of ERM. In particular, as discussed above, we show that when estimating treatment-specific hazard functions, *multiple* sources of covariate shift arise. (ii) We theoretically analyse their effects by adapting recent generalization bounds from domain adaptation and treatment effect estimation to our setting and discuss implications for model design. This analysis provides new insights that are of independent interest also in the context of hazard function estimation in the absence of treatments. (iii) Based on these insights, we propose a new model (SurvITE) relying on balanced representations that allows for estimation of treatment-specific target parameters (hazard and survival functions) in the survival context, as well as a sister model (SurvIHE), which can be used for individualized hazard estimation in standard survival settings (without treatments). We investigate performance across a range of experimental settings and empirically confirm that SurvITE outperforms a range of natural baselines by addressing covariate shifts from various sources.

## 2  Problem Definition

In this section, we discuss the problem setup of heterogeneous treatment effect estimation from time-to-event data, our target parameters and assumptions. In Appendix A, we present a self-contained introduction to and comparison with heterogeneous treatment effect estimation with standard (binary/continuous) outcomes.

**Problem setup.** Assume we observe a time-to-event dataset $\mathcal{D} = \{(a_i, x_i, \tilde{\tau}_i, \delta_i)\}_{i=1}^n$ comprising realizations of the tuple $(A, X, \tilde{T}, \Delta) \sim \mathbb{P}$ for $n$ patients. Here, $X \in \mathcal{X}$ and $A \in \{0, 1\}$ are random variables for a covariate vector describing patient characteristics and an indicator whether a binary treatment was administered at baseline, respectively. Let $T \in \mathcal{T}$ and $C \in \mathcal{T}$ denote random variables for the time-to-event and the time-to-censoring; here, events are usually *adverse*, e.g. progression/onset of disease or even death, and censoring indicates loss of follow-up for a patient. Then, the *observed* time-to-event outcomes of each patient are described by $\tilde{T} = \min(T, C)$ and $\Delta = \mathbb{1}(T \leq C)$, which indicate the time elapsed until either an event or censoring occurs and whether the event was observed or not, respectively. Throughout, we treat survival time as discrete[2] and the time horizon as finite with pre-defined maximum $t_{\max}$, so that the set of possible survival times is $\mathcal{T} = \{1, \cdots, t_{\max}\}$.

---

[2]Where necessary, discretization can be performed by transforming continuous-valued times into a set of contiguous time intervals, i.e., $T = \tau$ implies $T \in [t_\tau, t_\tau + \delta t)$ where $\delta t$ implies the temporal resolution.

We transform the *short* data structure outlined above to a so-called *long* data structure which can be used to *directly* estimate conditional hazard functions using standard machine learning methods [20]. We define two counting processes $N_T(t)$ and $N_C(t)$ which track events and censoring, i.e. $N_T(t) = \mathbb{1}(\tilde{T} \leq t, \Delta = 1)$ and $N_C(t) = \mathbb{1}(\tilde{T} \leq t, \Delta = 0)$ for $t \in \mathcal{T}$; both are zero until either an event or censoring occurs. By convention, we let $N_T(0) = N_C(0) = 0$. Further, let $Y(t) = \mathbb{1}(N_T(t) = 1 \cap N_T(t-1) = 0)$ be the indicator for an event occuring exactly at time $t$; thus, for an individual with $\tilde{T} = \tau$ and $\Delta = 1$, $Y(t) = 0$ for all $t \neq \tau$, and $Y(t) = 1$ at the event time $t = \tau$. The conditional hazard is the probability that an event occurs *at* time $\tau$ given that it does not occur before time $\tau$, hence it can be defined as [22]

$$\lambda(\tau|a, x) = \mathbb{P}(\tilde{T} = \tau, \Delta = 1|\tilde{T} \geq \tau, A = a, X = x) \\ = \mathbb{P}(Y(\tau) = 1|N_T(\tau-1) =_C (\tau-1) = 0, A = a, X = x) \tag{1}$$

It is easy to see from (1) that given data in long format, $\lambda(\tau|a, x)$ can be estimated for any $\tau$ by solving a standard classification problem with $Y(\tau)$ as target variable, considering only the samples *at-risk* at time $\tau$ in each treatment arm (individuals for which neither event nor censoring has occurred until that time point; i.e. the set $\mathcal{I}(\tau, a) \stackrel{\text{def}}{=} \{i \in [n] : N_T(\tau-1)_i = N_C(\tau-1)_i = 0 \cap A_i = a\}$). Finally, given the hazard, the associated survival function $S(\tau|a, x) = \mathbb{P}(T > \tau|A = a, X = x)$ can then be computed as $S(\tau|a, x) = \prod_{t \leq \tau} (1 - \lambda(t|a, x))$. The censoring hazard $\lambda_C(t|a, x)$ and survival function $S_C(t|a, x)$ can be defined analogously.

**Target parameters.** While the main interest in the standard treatment effect estimation setup with continuous outcomes usually lies in estimating only the (difference between) conditional outcome means under different treatments, there is a broader range of target parameters of interest in the time-to-event context, including both treatment-specific target functions and *contrasts* that represent some form of heterogeneous treatment effect (HTE). We define the treatment-specific (conditional) hazard and survival functions as

$$\lambda^a(\tau|x) = \mathbb{P}(T = \tau|T \geq \tau, do(A = a, C \geq \tau), X = x) \\ S^a(\tau|x) = \mathbb{P}(T > \tau|do(A = a, C \geq \tau), X = x) = \prod_{t \leq \tau} (1 - \lambda^a(t|x)) \tag{2}$$

Here, $do(\cdot)$ denotes [28]'s do-operator which indicates an intervention; in our context, $do(A = a, C \geq \tau)$ ensures that every individual is assigned treatment $a$ *and* is observed at (not censored before) the time-step of interest [20]. Below we discuss assumptions that are necessary to identify such interventional quantities from observational datasets in the presence of censoring.

Given $\lambda^a(\tau|x)$ and $S^a(\tau|x)$, possible HTEs of interest[3] include the difference in treatment-specific survival times at time $\tau$, i.e. $\text{HTE}_{surv}(\tau|x) = S^1(\tau|x) - S^0(\tau|x)$, the difference in restricted mean survival time (RMST) up to time $L$, i.e. $\text{HTE}_{rmst}(x) = \sum_{t_k \leq L} (S^1(t_k|x) - S^0(t_k|x)) \cdot (t_k - t_{k-1})$, and hazard ratios. In the following, we will focus on estimation of the treatment specific hazard functions $\{\lambda^a(t|x)\}_{a \in \{0,1\}, t \in \mathcal{T}}$ as this can be used to compute survival functions and causal contrasts.

**Assumptions.** (*1. Identification*) To identify interventional quantities from observational data, it is necessary to make a number of *untestable* assumptions on the underlying data-generating process (DGP) [28] – this generally limits the ability to make causal claims to settings where sufficient domain knowledge is available. Here, as [20, 21], we assume the data was generated from the fairly general directed acyclic graph (DAG) presented in Fig. 1. As there are no arrows originating in hidden nodes entering treatment or censoring nodes, this graph formalizes *(1.a) The 'No Hidden Confounders' Assumption* in static treatment effect estimation and *(1.b) The 'Censoring At*

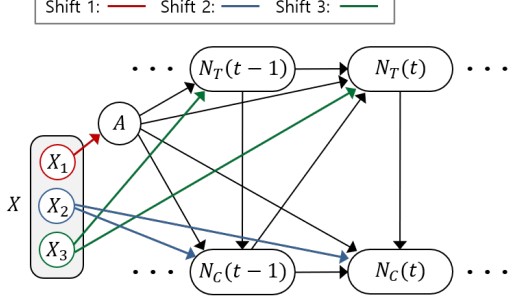

Figure 1: The assumed underlying DAG. Covariates $X$ can be split into (possibly overlapping) subsets $X_1$, $X_2$ and $X_3$, determining treatment selection, informative censoring, and event times, respectively.

---

[3]*Note:* All parameters of interest to us are *heterogeneous* (also sometimes referred to as *individualized*), i.e. a function of the covariates $X$, while the majority of existing literature in (bio)statistics considers *population average* parameters that are functions of quantities such as $\mathbb{P}(T > \tau|do(A = a))$, which average over all $X$.

*Random' Assumption* in survival analysis [20]. The latter is necessary here as estimating an effect of treatment on event time implicitly requires that censoring can be 'switched off' – i.e. intervened on. This graph implicitly also formalizes *(1.c) The Consistency Assumption*, i.e. that observed outcomes are 'potential' outcomes under the observed intervention, as each node in a DAG is *defined* as a function of its ancestors and exogenous noise [20]. Under these assumptions, $\lambda^a(\tau|x) = \lambda(\tau|a,x)$.

*(2. Estimation)* To enable nonparametric estimation of $\lambda^a(\tau|x)$ for some $\tau \in \mathcal{T}$, we additionally need to assume that the interventions of interest are observed with non-zero probability; within different literatures these assumptions are known under the label of 'overlap' or 'positivity' [9, 19]. In particular, for $0 < \epsilon_1, \epsilon_2, < 1$ we need that *(2.a)* $\epsilon_1 < \mathbb{P}(A = a|X = x) < 1 - \epsilon_1$, i.e. treatment assignment is non-deterministic, and that *(2.b)* $\mathbb{P}(N_C(t) = 0|A = a, X = x) > \epsilon_2$ for all $t < \tau$, i.e. no individual will be deterministically censored before $\tau$. Finally, because $\lambda^a(\tau|x)$ is a probability defined *conditional* on survival up to time $\tau$, we need to assume that *(2.c)* $\mathbb{P}(N_T(\tau-1) = 0|A = a, X = x) > \epsilon_3 > 0$ for it to be well-defined. We formally state and discuss all assumptions in more detail in Appendix C.

# 3 Challenges in Learning Treatment-Specific Hazard Functions using ERM

**Preliminaries: ERM under Covariate Shift.** Recall that in problems with covariate shift, the training distribution $X, Y \sim \mathbb{Q}_0(\cdot)$ used for ERM and target distribution $X, Y \sim \mathbb{Q}_1(\cdot)$ are mismatched: One assumes that the marginals do not match, i.e. $\mathbb{Q}_0(X) \neq \mathbb{Q}_1(X)$, while the conditionals remain the same, i.e. $\mathbb{Q}_0(Y|X) = \mathbb{Q}_1(Y|X)$ [29]. If the hypothesis class $\mathcal{H}$ used in ERM does not contain the truth (or in the presence of heavy regularization), this can lead to suboptimal hypothesis choice as $\arg\min_{h \in \mathcal{H}} \mathbb{E}_{X,Y \sim \mathbb{Q}_1(\cdot)}[\ell(Y, h(X))] \neq \arg\min_{h \in \mathcal{H}} \mathbb{E}_{X,Y \sim \mathbb{Q}_0(\cdot)}[\ell(Y, h(X))]$ in general.

## 3.1 Sources of Covariate Shift in Learning Treatment-Specific Hazard Functions

We now consider how to learn a treatment-specific hazard function $\lambda^a(\tau|x)$ from observational data using ERM. As detailed in Section 2, we exploit the long data format by realizing that $\lambda^a(\tau|x)$ can be estimated by solving a standard classification problem with $Y(\tau)$ as dependent variable and $X$ as covariates, using only the samples at risk with treatment status $a$, i.e. $\mathcal{I}(\tau, a)$, which corresponds to solving the empirical analogue of the problem

$$\hat{\lambda}^a(\tau|x) \in \arg\min_{h_{a,\tau} \in \mathcal{H}} \mathbb{E}_{X,Y(\tau) \sim \mathbb{P}_{a,\tau}(\cdot)}[\ell(Y(\tau), h_{a,\tau}(X)] \tag{3}$$

where we use $\mathbb{P}_{a,\tau}$ to refer to the observational (at-risk) distribution $\mathbb{P}_{a,\tau}(X, Y(\tau)) = \lambda_T^a(\tau|X)\mathbb{P}_{a,\tau}(X)$ with $\mathbb{P}_{a,\tau}(X) = \mathbb{P}(X|N_T(\tau-1) = N_C(\tau-1) = 0, A = a) = \mathbb{P}(X|\tilde{T} \geq \tau, A = a)$. If the loss function $\ell$ is chosen to be the log-loss, this corresponds to optimizing the likelihood of the hazard.

The observational (at-risk) covariate distribution $\mathbb{P}_{a,\tau}(X)$, however, is *not* our target distribution: instead, to obtain reliable hazard estimates for the whole population, we wish to optimize the fit over the population at baseline, i.e. the marginal distribution $X \sim \mathbb{P}(X)$ which we will refer to as $\mathbb{P}_0(X)$ below to emphasize it being the baseline at-risk distribution[4]. Here, differences between $\mathbb{P}_0(X)$ and the population at-risk $\mathbb{P}_{a,\tau}(X)$ can arise due to three distinct sources of covariate shift:

- *(Shift 1) Confounding/treatment selection bias*: if treatment is not assigned completely at random, then $\mathbb{P}(X|A = a) \neq \mathbb{P}_0(X)$ and the distribution of characteristics across the treatment arms differs already at baseline, thus $\mathbb{P}_{a,\tau}(X) \neq \mathbb{P}_0(X)$ in general.

- *(Shift 2) Censoring bias*: regardless of the presence of confounding, if the censoring hazard is not independent of covariates, i.e. $\lambda_C(\tau|a, x) \neq \lambda_C(\tau|a)$, then the population at-risk changes over time such that $\mathbb{P}_{a,\tau_1}(X) \neq \mathbb{P}_{a,\tau_2}(X) \neq \mathbb{P}_0(X)$ in general. If, in addition, there are differences between the treatment-specific censoring hazards, then the at-risk distribution will also differ across treatment arms at any given time-point, i.e. $\mathbb{P}_{a,\tau}(X) \neq \mathbb{P}_{1-a,\tau}(X)$ for $\tau > 1$ in general.

- *(Shift 3) Event-induced shifts*: Counterintuitively, even in the absence of both confounding and censoring, there will be covariate shift in the at-risk population if the event-hazard depends on covariates, i.e. if $\lambda(\tau|a, x) \neq \lambda(\tau|a)$ then $\mathbb{P}_{a,\tau_1}(X) \neq \mathbb{P}_{a,\tau_2}(X) \neq \mathbb{P}_0(X)$ in general. Further, if there are heterogenous treatment effects, then $\mathbb{P}_{a,\tau}(X) \neq \mathbb{P}_{1-a,\tau}(X)$ for $\tau > 1$ in general.

---

[4]With slight abuse of notation, we will use $\mathbb{P}_0$ and $\mathbb{P}_{a,\tau}$ also to refer to densities of continuous $x$

**What makes the survival treatment effect estimation problem unique?** While *Shift 1* arises also in the standard treatment effect estimation setting, *Shift 2* and *Shift 3* arise uniquely due to the nature of time-to-event data[5]. Thus, estimating treatment effects from time-to-event data is inherently more involved than estimating treatment effects in the standard static setup, as covariate shift at time horizon $\tau > 1$ can arise *even in a randomized control trial (RCT)*. Thus, in addition to the overall at-risk population changing over time, both treatment effect heterogeneity and treatment-dependent censoring can lead to differences in the composition of the population at-risk in each treatment arm. Further, Shifts 1, 2 and 3 can also interact to create more extreme shifts; e.g. if treatment selection is based on the same covariates as the event process (i.e. $X_1 = X_3$ in Fig. 1) then event-induced shift can amplify the selection effect over time (refer to Appendix E for a synthetic example of this).

### 3.2 Possible Remedies and Theoretical Analysis

A natural solution to tackle bias in ERM caused by covariate shift is to use importance weighting [30]; i.e. to reweight the empirical risk by the density ratio of target $\mathbb{P}_0(X)$ and observed distribution $\mathbb{P}_{a,\tau}(X)$. If we wanted to obtain a hazard estimator for $(\tau, a)$, optimized towards the marginal population, optimal importance weights are given by $w_{a,\tau}^*(x) = \frac{\mathbb{P}_0(x)}{\mathbb{P}_{a,\tau}(x)} = \frac{p_{\tau,a}}{e_a(x)r_a(x,\tau)}$ with $p_{\tau,a} = \mathbb{P}(\tilde{T} \geq \tau, A = a)$, $e_a(x) = \mathbb{P}(A = a|X = x)$ the propensity score, and $r^a(x,\tau) = \mathbb{P}(\tilde{T} \geq \tau|A = a, X = x)$ the probability to be at risk, i.e. the probability that neither event nor censoring occurred before time $\tau$. These weights are well-defined due to the overlap assumptions detailed in Sec. 2; however, they are in general unknown as they *depend on the unknown target parameters* $\lambda^a(\tau|x)$ through $r^a(x,\tau)$. Further, especially for large $\tau$, these weights might be very extreme even if known, which can lead to highly unstable results [31] – making biased yet stabilized weighting schemes, e.g. truncation, a good alternative. Therefore, we only assume access to some (possibly imperfect) weights $w_{a,\tau}(x)$ s.t. $\mathbb{E}_{X \sim \mathbb{P}_{a,\tau}}[w_{a,\tau}(x)] = 1$, so that we can create a weighted distribution $\mathbb{P}_{a,\tau}^w = w_{a,\tau}(x)\mathbb{P}_\tau^a(x)$. (Note: $\mathbb{P}_\tau^a(x)$ can be recovered by using $w_{a,\tau}(x) = 1$.)

Either instead of [8, 9] or in addition to weighting [10, 12, 14, 32], the literature on learning balanced representations for static treatment effect estimation has focused on finding a different remedy for distributional differences between treatment arms: creating representations $\Phi : \mathcal{X} \to \mathcal{R}$ which have similar (weighted) distributions across arms as measured by an integral probability metric (IPM), motivated by generalization bounds. As we show below, we can exploit a similar feature in our context by finding a representation that minimizes the IPM term not between treatment arms, but between covariate distribution at baseline $\mathbb{P}_0$ and $\mathbb{P}_{a,\tau}^w$. The proposition below bounds the target risk of a hazard estimator $\hat{\lambda}^a(\tau|x) = h(\Phi(x))$ relying on any representation. The proof, which relies on the concept of excess target information loss, proposed recently to analyze domain-adversarial training [33], and the standard IPM arguments made in e.g. [32], is stated in Appendix C.

**Proposition 1.** *For fixed $a, \tau$ and representation $\Phi : \mathcal{X} \to \mathcal{R}$, let $\mathbb{P}_0^\Phi$, $\mathbb{P}_{a,\tau}^\Phi$ and $\mathbb{P}_{a,\tau}^{w,\Phi}$ denote the target, observational, and weighted observational distribution of the representation $\Phi$. Define the pointwise losses*

$$\ell_{h,\mathbb{Q}}(x; a, \tau) \stackrel{\text{def}}{=} \mathbb{E}_{Y(\tau)|x,a \sim \mathbb{Q}}[\ell(Y(\tau), h(\Phi(X)))|X = x, A = a]$$

$$\ell_{h,\mathbb{Q}^\Phi}(\phi; a, \tau) \stackrel{\text{def}}{=} \mathbb{E}_{Y(\tau)|\phi,a \sim \mathbb{Q}^\Phi}[\ell(Y(\tau), h(\Phi))|\Phi = \phi, A = a]$$

(4)

*of (hazard) hypothesis $h \equiv h_{a,\tau} : \mathcal{R} \to [0, 1]$ w.r.t. distributions in covariate and representation space, respectively. Assume there exists a constant $C_\Phi > 0$ s.t. $C_\Phi^{-1}\ell_{h,\mathbb{P}_{a,\tau}^{w,\Phi}}(\phi, a, \tau) \in \mathcal{G}$ for some family of functions $\mathcal{G}$. Then we have that*

$$\underbrace{\mathbb{E}_{X \sim \mathbb{P}_0}[\ell_{h,\mathbb{P}}(X; a, \tau)]}_{\text{Target Risk}} \leq \underbrace{\mathbb{E}_{X \sim \mathbb{P}_{a,\tau}}[w_{a,\tau}(X)\ell_{h,\mathbb{P}}(X; a, \tau)]}_{\text{Weighted observational risk}} + C_\Phi \underbrace{IPM_G(\mathbb{P}_0^\Phi, \mathbb{P}_{a,\tau}^{w,\Phi})}_{\text{Distance in }\Phi\text{-space}} + \underbrace{\eta_\Phi^l(h)}_{\text{Info loss}}$$

(5)

---

[5]Interestingly, changes of the at-risk population over time arise also in standard survival problems (without treatments); yet in the context of *prediction* these do not matter: as the at-risk population at any time-step is also the population that will be encountered at test-time, this shift in population over time is not problematic, unless it is caused by censoring. If, however, our goal is *estimation* of the best target parameter (here: the hazard at a specific point in time $\tau$) over the whole population, this corresponds to a setting where the ideal evaluation is performed on a population different from the observed one – and hence requires careful consideration of the consequences of the covariate shifts discussed above.

*where IPM$_\mathcal{G}(\mathbb{P}, \mathbb{Q}) = \sup_{g \in \mathcal{G}} \left| \int g(x)(\mathbb{P}(x) - \mathbb{Q}(x))dx \right|$ and we define the excess target informa-tion loss $\eta_\Phi^\ell(h)$ analogously to [33] as $\eta_\Phi^\ell(h) \stackrel{\text{def}}{=} \mathbb{E}_{X \sim \mathbb{P}}[\xi_{\mathbb{P}_0^\Phi, \mathbb{P}}(X) - \xi_{\mathbb{P}_{a,\tau}^{w,\Phi}, \mathbb{P}}(X)]$ with $\xi_{\mathbb{Q}^\Phi, \mathbb{Q}}(x) \stackrel{\text{def}}{=} \ell_{h, \mathbb{Q}^\Phi}(\phi; a, \tau) - \ell_{h, \mathbb{Q}}(x; a, \tau)$. For invertible $\Phi$, $\eta_\Phi^\ell(h) = \xi_{\mathbb{Q}^\Phi, \mathbb{Q}}(x) = 0$.*

Unlike the bounds provided in [9, 10, 32, 14, 27], this bound does not rely on representations to be invertible; we consider this feature important as none of the works listed actually enforced invertibility in their proposed algorithms. Given bound (5), it is easy to see why non-invertibilty can be useful: for any (possibly non-invertible) representation for which it holds that $Y(\tau) \perp\!\!\!\perp X | \Phi(X), A$, it also holds that $\eta_\Phi^\ell(h) = \xi_{\mathbb{P}^\Phi, \mathbb{P}}(x) = \xi_{\mathbb{P}_{a,\tau}^{w,\Phi}, \mathbb{P}}(x) = 0$ and the causally identifying restrictions continue to hold. A simple representation for which this property holds is a selection mechanism that chooses only the causal parents of $Y(\tau)$ from within $X$; if $X$ can be partitioned into variables affecting the instantaneous risk ($X_3$ in Fig. 1), and variables affecting *only* treatment assignment ($X_1 \setminus X_3$) and/or censoring mechanism ($X_2 \setminus X_3$), then the IPM term can be reduced by a representation which drops the latter two sets of variables – or irrelevant variables correlated with any such variables – without affecting $\eta_\Phi^\ell(h)$. As a consequence, event-induced covariate shift can generally not be *fully* corrected for using non-invertible representations (unless the variables affecting event time are different at every time-step). Further, given perfect importance weights $w^*$, both $\eta_\Phi^\ell(h)$ and IPM term are zero.

Except for the dependence on $\eta_\Phi^\ell(h)$, this bound differs from the regression-based bound for survival treatment effects stated in [27] (which is identical to the original treatment effect bound in [9]) in that we have dependence on $\tau$ in the IPM term, which, among other things, explicitly captures the effect of censoring. Our bound motivates that, instead of finding representations that balance treatment- and control group at baseline (or at each time step) we should find representations that balance $\mathbb{P}_{a,\tau}^\Phi$ towards the *baseline distribution* $\mathbb{P}_0^\Phi$ for each time step, which motivates our method detailed below. If, instead, we would apply the IPM-term to encourage only the arm-specific at-risk distributions at each time-step to be similar, this would correct only for shifts due to (i) confounding at baseline, (ii) treatment-induced differences in censoring and (iii) treatment-induced differences in events. It will, however, not allow handling the event- and censoring-induced shifts that occur regardless of treatment status. Note that this bound therefore also motivates the use of balanced representations for modeling time-to-event outcomes in the presence of informative censoring even in the standard prediction setting, which is a finding that could be of independent interest for the ML survival analysis literature.

### 3.3 From hazards to survival functions

If the ultimate goal is to use the hazard function to estimate survival functions as $\hat{S}^a(\tau|x) = \prod_{t \leq \tau} \left(1 - \hat{\lambda}^a(t|x)\right)$, the best target population to consider during hazard estimation may not be the marginal distribution. Instead, the optimal target population may depend on the metric by which we wish to evaluate the resulting survival function. If we wanted to find the survival function that maximises the complete data likelihood (corresponding to the hypothetical setting in which we intervened to set $A = a$ and $C \geq \tau$), the target population (at each time step $t$) would be $\mathbb{P}(X | T \geq t, do(A = a, C \geq t))$ – the population that preserves event-induced shift but removes selection- and censoring-induced shifts. If, instead, we focused on the MSE of estimating the survival function (as in our experiments), it becomes more difficult to derive an exact target population for estimating the hazards. If we assume access to a perfect estimate of the survival function for the first $\tau - 1$ time steps (i.e. $\hat{S}(\tau - 1|x) = S(\tau - 1|x)$) and focus only on estimating the next hazard, $\lambda^a(\tau|X)$, we can write

$$\mathbb{E}_{X \sim \mathbb{P}_0}[(S^a(\tau|X) - \hat{S}^a(\tau|X))^2] = \mathbb{E}_{X \sim \mathbb{P}_0}[(S^a(\tau-1|X)(1-\lambda^a(\tau|X)) - \hat{S}^a(\tau-1|X)(1-\hat{\lambda}^a(\tau|X)))^2]$$
$$= \mathbb{E}_{X \sim \mathbb{P}_0}[S^a(\tau-1|X)^2(\hat{\lambda}^a(\tau|X) - \lambda^a(\tau|X))^2]$$

and notice that the MSE will implicitly down-weigh individuals with lower survival probability[6].

Defining an exact target population for the hazard when the goal is to also estimate the survival function well is thus not straightforward, making exact importance weighting difficult. Additionally, unlike the marginal population, interventional populations which change over time, such as $\mathbb{P}(X | T \geq t, do(A = a, C \geq t))$, are never observed in practice and hence cannot be used to perform balancing

---

[6]Due to the square in the term $S^a(\tau - 1|X)^2$, this will be even more extreme than exact up-weighting of the population $\mathbb{P}(X | T \geq \tau, do(A = a, C \geq \tau))$.

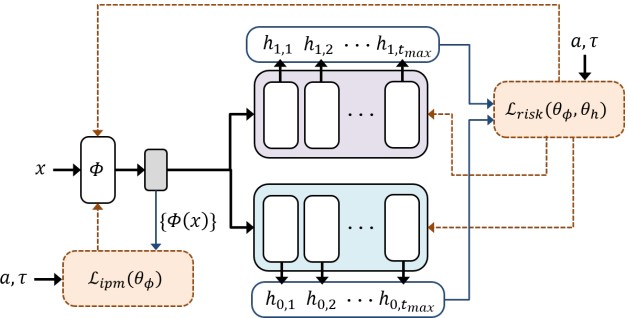

Figure 2: SurvITE architecture.

regularization of a representation using empirical IPM-terms. Therefore, we refrain from using importance weighting in our method (which is described in the next section), and resort to using the marginal population for balancing regularization of the representation throughout. Intuitively, as outlined in the previous section, we expect that doing so will *not* over-correct for event-induced shifts that are predictive of outcome (and should hence be preserved) as such "over-balancing" would reduce the predictive power of the representation, which would immediately be penalized by the presence of the expected loss component in the bound[7]. Additionally, we expect that using the marginal population for balancing could be useful also for estimating the survival function even in the absence of selection- and censoring-induced shifts, as it may help to remove the effect of variables that appear spuriously correlated with outcome over time.

## 4 SurvITE: Estimating HTEs from Time-to-Event Data

Based on the theoretical analysis above, we propose a novel deep learning approach to HTE estimation from observed time-to-event data, which we refer to as SurvITE (Individualized Treatment Effect estimator for Survival analysis).[8] The network architecture is illustrated in Figure 2. Note that even in the absence of treatments we can use this architecture for estimation of hazards and survival functions by using only one treatment $a = 0$. As we show in the experiments, this version of our method – SurvIHE (Individualized Hazard Estimator for Survival analysis) – is of independent interest in the standard survival setting, as it tackles Shifts 2 & 3. Below, we describe the empirical loss functions we use to find representation $\Phi$ and hypotheses $h_{a,\tau}$.

Let $\Phi : \mathcal{X} \to \mathcal{R}$ denote the *representation* (parameterized by $\theta_\phi$) and $h_{a,\tau} : \mathcal{R} \to [0,1]$ the *hazard estimator* for treatment $a$ and time $\tau$ (parameterized by $\theta_{h_{a,\tau}}$), each implemented as a fully-connected neural network. While the output heads are thus unique to each treatment-group time-step combination, we allow hazard estimators to share information by using *one* shared representation for all hazard functions. This allows for both borrowing of information across different $a, \tau$ and significantly reduces the number of parameters of the network. Then, given the time-to-event data $\mathcal{D}$, we use the following empirical loss functions for the observational risk and the IPM term:

$$\mathcal{L}_{risk}(\theta_\phi, \theta_h) = \frac{1}{t_{\max}} \sum_{t=1}^{t_{\max}} \sum_{i:\tilde{\tau}_i \geq t} n_{1,t}^{-1} a_i \ell\big(y_i(t), h_{1,t}(\Phi(x_i))\big) + n_{0,t}^{-1}(1-a_i)\ell\big(y_i(t), h_{0,t}(\Phi(x_i))\big),$$

$$\mathcal{L}_{ipm}(\theta_\phi) = \sum_{a \in \{0,1\}} \sum_{t=1}^{t_{\max}} Wass\big(\{\Phi(x_i)\}_{i=1}^n, \{\Phi(x_i)\}_{i:\tilde{\tau}_i \geq t, a_i = a}\big),$$

where $Wass(\cdot, \cdot)$ is the finite-sample Wasserstein distance [34]; refer to Appendix D for further detail. Note that $\mathcal{L}_{ipm}(\theta_\phi)$, which penalizes the discrepancy between the baseline distribution and *each* at-risk distribution $\mathbb{P}_{a,\tau}^\Phi$, simultaneously tackles all three sources of shifts. Further, $n_{a,t} = |\mathcal{I}(\tau, a)|$ is the number of samples at-risk in each treatment arm; its presence ensures that each $a, \tau$-combination contributes equally to the loss. Overall, we can find $\Phi$ and $h_{a,\tau}$'s that optimally trade off balance and

---

[7]In practice, we ensure this by weighting the contribution of the IPM term by a hyperparameter that is chosen to preserve predictive performance of the representation (see Appendix D).

[8]The source code for SurvITE is available in `https://github.com/chl8856/survITE`.

predictive power as suggested by the generalization bound (5) by minimizing the following loss:

$$\mathcal{L}_{target}(\theta_\phi, \theta_h) = \mathcal{L}_{risk}(\theta_\phi, \theta_h) + \beta \mathcal{L}_{ipm}(\theta_\phi) \tag{6}$$

where $\theta_h = \{\theta_{h_{a,\tau}}\}_{a \in \{0,1\}, \tau \in \mathcal{T}}$, and $\beta > 0$ is a hyper-parameter. The pseudo-code of SurvITE, the details of how to obtain $Wass(\cdot, \cdot)$ and how we set $\beta$ can be found in Appendix D.

## 5 Related work

**Heterogeneous treatment effect estimation (non-survival)** has been studied in great detail in the recent ML literature. While early work built mainly on tree-based methods [1, 2, 3, 4], many other methods, such as Gaussian processes [6, 7] and GANS [35], have been adapted to estimate HTEs. Arguably the largest stream of work [8, 9, 10, 11, 12, 13, 14, 15] built on NNs, due to their flexibility and ease of manipulating loss functions, which allows for easy incorporation of balanced representation learning as proposed in [8, 9] and motivated also the approach taken in this paper. Another popular approach has been to consider model-agnostic (or 'meta-learner' [36]) strategies, which provide a 'recipe' for estimating HTEs using *any* predictive ML method [36, 37, 38, 15]. Because of their simplicity, the *single model* (S-learner) – which uses the treatment indicator as an additional covariate in otherwise standard model-fitting – and *two model* (T-learner) – which splits the sample by treatment status and fit two separate models – strategies [36], can be directly applied to the survival setting by relying on a standard survival (prediction) method as base-learner.

**ML methods for survival prediction** continue to multiply; here we focus on the most related class of methods – namely on those nonparametrically modeling conditional hazard or survival functions – and *not* on those relying on flexible implementations of the Cox proportional hazards model (e.g. [39, 40, 41]) or modeling (log-)time as a regression problem (e.g. [42, 43, 44, 45, 46, 47]). One popular nonparametric estimator of survival functions is [48]'s random survival forest, which relies on the Nelson-Aalen estimator to nonparametrically estimate the cumulative hazard within tree-leaves. The idea of modeling discrete-time hazards directly using *any arbitrary classifier* and long data-structures goes back to at least [49], with implementations using NN-based methods presented in e.g. [50, 51, 52, 53]. [54] models the probability mass function instead of the hazard, and [55] use labels $\mathbb{1}\{T > t\}_{t \in \mathcal{T}}$ to estimate the survival function directly using multi-task logistic regression. For a more detailed overview of different strategies for estimating survival functions, refer to Appendix B.

**Estimating HTEs from time-to-event data** has been studied in much less detail. [23, 25] use tree-based nearest-neighbor estimates to estimate expected differences in survival time directly, and [24] use a BART-based S-learner to output expected differences in log-survival time. [56] performed a simulation study using different survival prediction models as base-learners for a two-model approach to estimating the difference in median survival time. Based on ideas from the semi-parametric efficiency literature, [26] and [57] propose estimators that target the (restricted) mean survival time *directly* and consequently *do not* output estimates of the treatment-specific hazard or survival functions. We consider the ability to output treatment-specific predictions an important feature of a model if the goal is to use model output to give decision support, given that it allows the decision-maker to trade-off relative improvement with the baseline risk of a patient. Finally, [27] recently proposed a generative model for treatment-specific event times which relies on balancing representations to balance only the treatment groups at baseline. This model does not output hazard-or survival functions, but can provide approximations by performing Monte-Carlo sampling.

## 6 Experiments

Unfortunately, when the goal is *estimating* (differences of) survival functions (instead of *predicting* survival), evaluation on real data will not reflect performance w.r.t. the intended baseline population. Therefore, we conduct a range of synthetic experiments with *known* ground truth. We evaluate the effects of different shifts separately by starting with survival estimation *without* treatments, and then introduce treatments. Finally, we use the real-world dataset Twins [58] which has uncensored survival outcomes for twins (where the treatment is 'being born heavier'), and is hence free of Shifts 1 & 2.

**Baselines.** Throughout, we use Cox regression (**Cox**), a model using a separate logistic regression to solve the hazard classification problem at each time-step (**LR-sep**), random survival forest (**RSF**), and a deep learning-based time-to-event method [54] (**DeepHit**) as natural baselines; when there

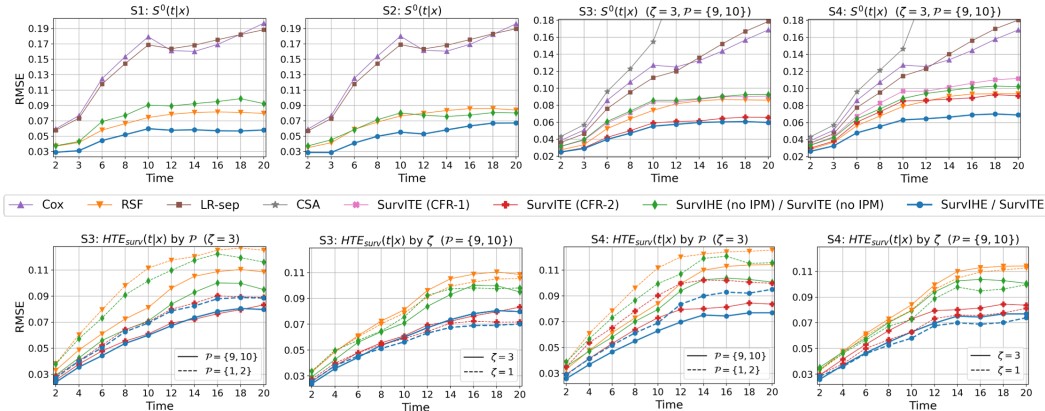

Figure 3: RMSE of estimating the survival function $S^0(t|x)$ (top) and the treatment effect $HTE_{surv}(t|x)$ (bottom) for different time steps across synthetic settings. Averaged across 5 runs.

are treatments, we use them in a two-model (T-learner) approach. In settings with treatments, we additionally use the CSA-INFO model of [27] (**CSA**), where we use its generative capabilities to approximate target quantities via monte-carlo sampling. Finally, we consider ablations of SurvITE (and SurvIHE); in addition to removing the IPM term (**SurvITE (no IPM)**), we consider two variants of SurvITE based on [9]'s CFRNet balancing term: **SurvITE (CFR-1)** creates a representation balancing treatment groups at baseline only, and **SurvITE (CFR-2)** creates a representation optimizing for balance of treatment groups *at each time step* (i.e. no balancing towards $\mathbb{P}_0$). We discuss implementation in Appendix D.

**Synthetic Experiments.** We consider a range of synthetic simulation setups (S1-S4) to highlight and isolate the effects of the different types of covariate shift. As event and censoring processes, we use

$$\lambda^a(t|x) = \begin{cases} 0.1\sigma(-5x_1^2 - a \cdot (\mathbb{1}\{x_3 \geq 0\} + 0.5)) & \text{for } t \leq 10 \\ 0.1\sigma(10x_2 - a \cdot (\mathbb{1}\{x_3 \geq 0\} + 0.5))) & \text{for } t > 10 \end{cases}, \quad \lambda_C(t|x) = 0.01\sigma(10x_4^2)$$

with treatment assignment mechanism $a \sim \text{Bern}(\xi \cdot \sigma(\sum_{p \in \mathcal{P}} x_p))$, with $\sigma$ the sigmoid function. Additionally, we assume administrative censoring at $t = 30$ throughout, i.e., $\lambda_C(30|x) = 1$, marking e.g. the end of a hypothetical clinical study. Covariates are generated from a 10-dimensional multivariate normal distribution with correlations, i.e. $X \sim \mathcal{N}(\mathbf{0}, \mathbf{\Sigma})$ where $\mathbf{\Sigma} = (1 - \rho)\mathbf{I} + \rho\mathbf{1}\mathbf{1}^\top$ with $\rho = 0.2$. We use 5000 independently generated samples each for training and testing.

In S1, we begin with the simplest case – *no* treatments and *no* censoring – using only $\lambda^0(t|x)$ to generate events, considering only event-induced shift (Shift 3). In S2, we introduce informative censoring using $\lambda_C(t|x)$ (Shift 2+3). In S3, we use treatments and consider biased treatment assignment (without censoring) (Shift 1+3). In S4, we consider the most difficult case with all three types of shift (Shift 1+2+3). In the latter two settings, we vary treatment selection by changing (i) whether the covariate set overlaps with the event-inducing covariates ($\mathcal{P}=\{1, 2\}$) or not ($\mathcal{P}=\{9, 10\}$) and (ii) the selection strength $\xi \in \{1, 3\}$. We present exploratory plots of these DGPs in Appendix E.

Fig. 3 (top) shows performance on estimating $S^0(t|x) = \prod_{k \leq t} \left(1 - \lambda^0(k|x)\right)$ for all scenarios and methods, while Fig. 3 (bottom) shows performance on estimating the difference in survival functions ($HTE_{surv}(t|x)$) for a selection of methods (for readability, full results in Appendix F). In Table 1, we further evaluate the estimation of differences in RMST (HTE$_{rmst}(x)$). Results for hazard estimation and additional performance metrics for survival *prediction* are reported in Appendix F. We observe that SurvITE (/SurvIHE) performs best throughout, and that introduction of the IPM term leads to substantial improvements across all scenarios. In S1 with only event-induced covariate shift and in S3/4 when treatment selection and event-inducing covariates overlap ($\mathcal{P}=\{1, 2\}$), balancing cannot remove all shift as the shift-inducing covariates are predictive of outcome; however, even here the IPM-term helps as it encourages dropping other covariates (which appear imbalanced due to correlations in $X$). While our method was motivated by theory for estimation of hazard functions, it thus indeed also leads to gains in survival function estimation. As expected, both Cox and LR-sep

Table 1: RMSE on estimation of $\text{HTE}_{rmst}(x)$ (mean $\pm$ 95%-CI) for different times for the Synthetic and Twins datasets ($L$s are the 25 & 75th and 75 & 95th percentiles of event times, respectively).

| Methods | S3 ($\zeta = 3$, no overlap) | | S4 ($\zeta = 3$, no overlap) | | Twins (no censoring) | | Twins (censoring) | |
|---|---|---|---|---|---|---|---|---|
| | $L = 10$ | $L = 20$ | $L = 10$ | $L = 20$ | $L = 30$ | $L = 180$ | $L = 30$ | $L = 180$ |
| Cox | 0.434±0.03 | 1.073±0.05 | 0.424±0.02 | 1.047±0.04 | 2.85±0.10 | 20.33±0.50 | 2.88±0.09 | 20.60±0.50 |
| RSF | 0.328±0.02 | 1.027±0.03 | 0.332±0.02 | 1.058±0.03 | 3.15±0.07 | 22.42±0.36 | 3.18±0.08 | 22.62±0.46 |
| LR-sep | 0.412±0.02 | 1.111±0.07 | 0.418±0.02 | 1.149±0.04 | 2.94±0.10 | 20.60±0.53 | 2.94±0.10 | 20.66±0.52 |
| DeepHit | 0.347±0.03 | 0.821±0.07 | 0.361±0.08 | 0.830±0.15 | 2.95±0.28 | 20.89±1.91 | 2.86±0.09 | 20.69±0.52 |
| CSA | 0.421±0.01 | 2.098±0.26 | 0.406±0.01 | 1.932±0.12 | 3.42±0.12 | 26.20±1.21 | 4.41±0.54 | 47.79±1.55 |
| SurvITE (no IPM) | 0.275±0.04 | 0.843±0.11 | 0.310±0.05 | 0.930±0.11 | 2.80±0.10 | 19.80±1.01 | 2.85±0.22 | 20.00±1.07 |
| SurvITE (CFR-1) | 0.269±0.04 | 0.825±0.09 | 0.341±0.02 | 1.016±0.10 | 2.68±0.06 | 19.16±0.37 | 2.67±0.15 | 19.10±0.85 |
| SurvITE (CFR-2) | 0.236±0.04 | 0.691±0.08 | 0.294±0.07 | 0.815±0.15 | 2.61±0.12 | 18.69±0.64 | 2.69±0.22 | 19.20±1.44 |
| **SurvITE** | **0.225±0.03** | **0.687±0.08** | **0.237±0.03** | **0.703±0.06** | **2.53±0.09** | **18.34±0.70** | **2.63±0.10** | **18.76±0.56** |

do not perform well as they are misspecified, while the nonparametric RSF is sufficiently flexible to capture the underlying DGP and usually performs similarly to SurvITE (architecture only), but is outperformed once the IPM term is added. For readability, we did not include DeepHit in Fig. 3; using table F.1 presented in Appendix F, we observe that DeepHit performs worse than the SurvITE architecture without IPM term, indicating that our model architecture alone is better suited for estimation of treatment-specific survival functions (note that [54] focused mainly on discriminative (predictive) performance, and not on the estimation of the survival function itself). Therefore, upon addition of the IPM-terms, the performance gap between SurvITE and DeepHit only becomes larger.

A comparison with ablated versions highlights the effect of using the baseline population to define balance; naive balancing across treatment arms (either at baseline – SurvITE(CFR-1), or over time – SurvITE(CFR-2)) is not as effective as using the baseline population as a target, especially at the later time steps where the effects of time-varying shifts worsen. While SurvITE(CFR-2) almost matches the performance of the full SurvITE in S3, it performs considerably worse in S4, indicating that this form of balancing suffers mainly due to its ignorance of censoring. Finally, a comparison with CSA highlights the value of modeling hazard functions directly: we found that Monte-Carlo approximation of the survival function using the generated event times gives very badly calibrated survival curves as event times generated by CSA were concentrated in a very narrow interval, leading to survival estimates of 0 and 1 elsewhere. Its performance on estimation of RMST was likewise poor; we conjecture that this is due to (i) CSA modeling continuous time, while the outcomes were generated using a coarse discrete time model, and (ii) the significant presence of administrative censoring.

**Real data: Twins.** Finally, we consider the Twins benchmark dataset, containing survival times (in days, administratively censored at t=365) of 11400 pairs of twins, which is used in [58, 35] to measure HTEs of birthweight on infant mortality. We split the data 50/50 for training and testing (by twin pairs), and similar to [35], use a covariate-based sampling mechanism to select only one twin for training to emulate selection bias. Further, we consider a second setting where we additionally introduce covariate-dependent censoring. For all discrete-time models, we use a non-uniform discretization to construct classification tasks because most events are concentrated in the first weeks. A more detailed description of the data and experimental setup can be found in Appendix E. As the data is real and ground truth probabilities are unknown, $\text{HTE}_{rmst}(x)$ is suited best to evaluate performance on estimating effect heterogeneity. The results presented in Table 1 largely confirm our findings on relative performance in the synthetic experiments; only RSF performs relatively worse on this dataset.

## 7 Conclusion

We studied the problem of inferring heterogeneous treatment effects from time-to-event data by focusing on the challenges inherent to treatment-specific hazard estimation. We found that a variety of covariate shifts play a role in this context, theoretically analysed their impact, and demonstrated across a range of experiments that our proposed method SurvITE successfully mitigates them.

**Limitations.** Like all methods for inferring causal effects from observational data, SurvITE relies on a set of strong assumptions which should be evaluated by a domain expert prior to deployment in practice. Here, the time-to-event nature of our problem adds an additional assumption ('random censoring') to the standard 'no hidden confounders' assumption in classical treatment effect estimation. If such assumptions are not properly assessed in practice, any causal conclusions may be misleading.

## Acknowledgments and Disclosure of Funding

We thank anonymous reviewers as well as members of the vanderschaar-lab for many insightful comments and suggestions. AC gratefully acknowledges funding from AstraZeneca. CL was supported through the IITP grant funded by the Korea government(MSIT) (No. 2021-0-01341, AI Graduate School Program, CAU). Additionally, MvdS received funding from the Office of Naval Research (ONR) and the National Science Foundation (NSF, grant number 1722516).

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
