# SurvITE: Learning Heterogeneous Treatment Effects from Time-to-Event Data
# Appendix

**Alicia Curth**[*]
University of Cambridge
amc253@cam.ac.uk

**Changhee Lee**[*]
Chung-Ang University
chl8856@gmail.com

**Mihaela van der Schaar**
University of Cambridge
University of California, Los Angeles
The Alan Turing Institute
mv472@cam.ac.uk

This appendix is organized as follows: We first present an extended overview of the standard treatment effect estimation setup and discuss differences with the time-to-event setting (Appendix A). Then, we give an extended review of strategies for nonparametric estimation of survival dynamics (Appendix B). In Appendix C we discuss technical details – assumptions and proofs – and Appendix D we discuss implementation. Appendix E contains additional descriptions of datasets and experimental setup and Appendix F presents additional results. Appendix G contains the NeurIPS checklist.

## A    Preliminaries on treatment effect estimation

In the standard treatment effect estimation setup with binary or continuous outcomes (see e.g. [1, 2, 3]), one usually observes a dataset $\mathcal{D} = \{(a_i, x_i, y_i)\}_{i=1}^n$ comprising $n$ realizations of the tuple $(A, X, Y)$. $X \in \mathcal{X}$ and $A \in \{0, 1\}$ represent patient characteristics and treatment assignment as in the main text. $Y \in \mathcal{Y}$ is usually a binary ($\mathcal{Y} = \{0, 1\}$) or continuous ($\mathcal{Y} = \mathbb{R}$) outcome. The target parameter of interest is often the *conditional average treatment effect (CATE)*

$$\tau(x) = \mathbb{E}[Y|X = x, do(A = 1)] - \mathbb{E}[Y|X = x, do(A = 0)] \tag{1}$$

which is impossible to estimate from observational data without further assumptions, as – due to the *fundamental problem of causal inference* [4] – every individual is only ever observed under *one* of the two possible interventions. CATE can therefore only be nonparametrically estimated under the imposition of untestable assumptions; here we rely on the standard ignorability assumptions [5] of *No hidden confounders (1.a)*, *Consistency (1.c)* and *Positivity/Overlap in treatment assignment (2.a)*.

### A.1    Comparison with the time-to-event treatment effects setup

The time-to-event setting is made more involved by (i) the presence of censoring and (ii) the interest in the *dynamics* of the underlying survival process.

Censoring – the removal of some individuals from the sample before having observed their event time – further complicates the treatment effect estimation problem, because every individual's outcome (time-to-event) is now observed under *at most* one intervention. The presence of censoring adds an additional source of covariate shift, and the need to rely on the assumptions of *Censoring at random (1.b)* and *Positivity in censoring (2.b)*. Censoring is, however, different from complete missingness of the outcome as the censoring time provides *some* information on the outcome – an individual has survived *at least* until the censoring time.

While the difference in expected survival time (the time-to-event equivalent in CATE) can be the treatment effect of interest in a study, many survival analysis problems are concerned with target parameters that capture differences in the *dynamics* of the underlying survival process across

---

[*]Equal contribution

35th Conference on Neural Information Processing Systems (NeurIPS 2021).

treatments, e.g. hazard ratios or differences in survival functions – which substantially increases the number of possible target parameters to model (beyond 'only' CATE). Instead of only modeling expected outcomes (as would be the case in the standard setup as discussed above), modeling survival dynamics through e.g. the treatment-specific hazard function can therefore often be of interest. Nonparametrically modeling hazard functions introduces the additional assumption on *Positivity of events (2.c)*.

## B    Strategies for loss-based discrete-time hazard and survival function estimation

In this section, we review strategies for nonparametric (or machine-learning based) estimation of the dynamics underlying discrete-time event processes. Here, we consider on the standard case *without treatments* to highlight how a dependence on different populations arises in different modeling strategies, and follow closely the exposition of different strategies in [6]. We focus on loss functions that can be used for implementation to highlight that these approaches are valid for use of *any* classifier, and then briefly mention specific instantiations of such approaches from related work.

**Preliminaries.** In addition to hazard and survival function defined in the main text, define the probability mass functions (PMF) as

$$f(\tau|x) = \mathbb{P}(T = \tau|X = x) \text{ and } f_C(\tau|x) = \mathbb{P}(C = \tau|X = x) \tag{B.1}$$

Note that a hazard $\lambda(\tau|x) = \mathbb{P}(T = \tau|T \leq \tau, X = x)$ can then also be defined as $\lambda(\tau|x) = \frac{f(\tau|x)}{S(\tau-1|x)}$. Further, recall that the survival function $S(\tau|x) = \prod_{t \leq \tau} \big(1 - \lambda(t|x)\big)$, so that the PMF can be rewritten as $f(\tau|x) = \lambda(\tau|x)S(\tau - 1|x) = \lambda(\tau|x) \prod_{t \leq \tau-1} \big(1 - \lambda(t|x)\big)$.

### B.1    Likelihood-based hazard estimation

Under the assumption of random censoring (which is discussed further in Appendix C.1), the likelihood function of the observed (short) data factorizes; i.e.

$$
\begin{aligned}
\mathbb{P}(\tilde{T} = \tilde{\tau}, \Delta = \delta|X = x) &= \mathbb{P}(T = \tilde{\tau}, C \geq \tilde{\tau}|X = x)^\delta \mathbb{P}(T > \tilde{\tau}, C = \tilde{\tau}|X = x)^{1-\delta} \\
&= \big[\mathbb{P}(T = \tilde{\tau}|X = x)\mathbb{P}(C \geq \tilde{\tau}|X = x)\big]^\delta \big[\mathbb{P}(T > \tilde{\tau}|X = x)\mathbb{P}(C = \tilde{\tau}|X = x)\big]^{1-\delta} \\
&= \big[f(\tilde{\tau}|x)(S_C(\tilde{\tau}|x) + f_C(\tilde{\tau}|x))\big]^\delta \big[S(\tilde{\tau}|x)f_C(\tilde{\tau}|x))\big]^{1-\delta} \\
&= \underbrace{f(\tilde{\tau}|x)^\delta S(\tilde{\tau}|x)^{1-\delta}}_{\text{Event-relevant}} \underbrace{f_C(\tilde{\tau}|x)^{1-\delta}(S_C(\tilde{\tau}|x) + f_C(\tilde{\tau}|x))^\delta}_{\text{Ignorable censoring mechanism}}
\end{aligned}
$$

By the likelihood principle, the parts pertaining to censoring are *ignorable*, hence we can consider censoring and event likelihoods separately [7]. The likelihood contribution of observation $i$ to the negative time-to-event likelihood can then be written as:

$$L_i = -f(\tilde{\tau}_i|x_i)^{\delta_i} S(\tilde{\tau}_i|x_i)^{1-\delta_i} = \lambda(\tilde{\tau}_i|x)^{\delta_i}(1 - \lambda(\tilde{\tau}_i|x))^{1-\delta_i} \prod_{t \leq \tilde{\tau}_i-1} \big(1 - \lambda(t|x)\big) \tag{B.2}$$

so that, after taking the logarithm and summing over all $i \in [n]$ we have that

$$
\begin{aligned}
\mathcal{L} &= -\sum_{i=1}^n \Big(\delta_i log(\lambda(\tilde{\tau}_i|x)) + (1 - \delta_i)log(1 - \lambda(\tilde{\tau}_i|x)) + \sum_{t \leq \tilde{\tau}_i-1} log(1 - \lambda(t|x)))\Big) \\
&= -\sum_{t=1}^{t_{max}} \sum_{i=1}^n \mathbb{1}(\tilde{\tau}_i \geq t)\big(y_i(t)log(\lambda(t|x)) + (1 - y_i(t))log(1 - \lambda(t|x)))\big)
\end{aligned} \tag{B.3}
$$

with $y_i(t) = \mathbb{1}(\tilde{\tau}_i = t, \delta_i = 1) = \mathbb{1}(N_T(t)_i = 1 \cap N_T(t-1)_i = 0)$ as in the main text. Thus, the classification approach with log-loss is *equivalent* to optimizing for the likelihood of the hazard. Optimizing the likelihood of the hazard thus suffers from the exact same shifts as the classification approach, namely the shifts induced by focusing on the 'at-risk' population at any time-step: the log-loss also has dependence on $\mathbb{1}(\tau_i \geq t)$. Note that, as we illustrate in section B.1.1, under such

shifts, optimizing the likelihood is only problematic if the model for $\lambda(\tau|x)$ is misspecified – a well-established fact in the literature on covariate shift [8].

Depending how $\lambda(\tau|x)$ is parameterized, different models proposed in related work arise. The idea to use a classification approach dates back to at least the logistic-hazard model in [9], and is reviewed in more detail in [10]. The first NN-based implementation that we are aware of is [11], which parameterizes $\lambda(\tau|x)$ by using one shared network for all $\tau \in \mathcal{T}$ where the time-indicator $\tau$ is passed as an additional covariate. [12] instead propose a network with some shared layers and $\tau$-specific output layers (resulting in a model similar to the SurvIHE base-model). Finally, [13]'s DSRA parameterizes $\lambda(\tau|x)$ using a recurrent network which encodes the structure shared across time.

### B.1.1   Illustration: Why (mis)specification matters

To briefly illustrate when event-induced at-risk population shift matters, we consider two simple toy examples: we rely on event-processes with covariate-dependent but time-constant hazards, i.e. $\lambda(\tau_1|x) = \lambda(\tau_2|x)$, and there are 5 multivariate normal correlated covariates, of which only $X_1$ determines the hazard. We parameterize hazard estimators using a separate logistic regression at each time step $t$. We consider one process where this logistic regression is correctly specified for the underlying hazard function, as $\lambda_1(\tau|x) = \sigma(x_1 - 0.25)$. We consider another process where this logistic regression is misspecified, as $\lambda_2(\tau|x) = \sigma(\mathbb{1}(x_1 > 0)x_1 - 0.25)$ (i.e. there is a nonlinearity that cannot be perfectly captured by a simple logistic regression).

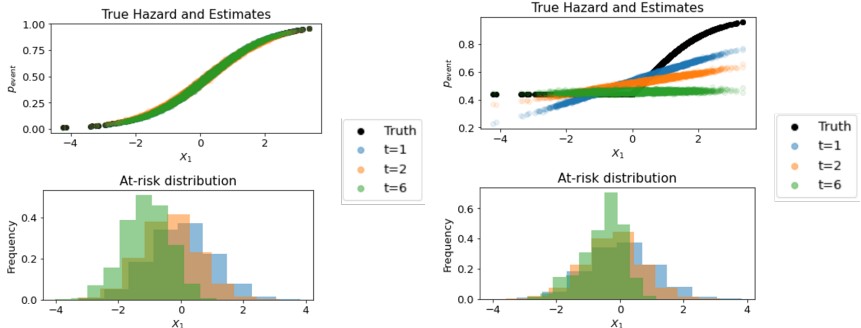

(a) Hazard estimates and at-risk distribution for $\lambda_1(\tau|x)$ (well-specified model)

(b) Hazard estimates and at-risk distribution for $\lambda_2(\tau|x)$ (misspecified model)

Figure B.1: Toy example highlighting that covariate shift plays no role when models are well-specified (left) but matters under misspecification (right).

As can be seen in Fig. B.1, both processes lead to event-induced covariate shift. However, this shift has no effect on hazard estimator performance over time when the model is correctly specified. Yet, when the model is incorrectly specified, the estimator has to trade off making errors in different regions of the covariate space. The optimal trade-off w.r.t. the baseline distribution is made by the hazard classifier at $t = 1$ where the at-risk distribution corresponds to the marginal distribution of covariates. Due to event-induced covariate shift, hazard estimates become increasingly biased towards the survivor population at later time-steps.

### B.2   Survival-based estimation

An alternative approach to targeting the likelihood of the hazard would be to target the survival function directly by realizing that $\mathbb{P}(T > t|X = x) = \mathbb{E}[\mathbb{1}(T > t)|X = x]$, so that the survival function can be estimated directly by solving $t_{\max}$ classification problems with targets $\{\mathbb{1}(T > t)\}_{t \in \mathcal{T}}$. This considers a loss function

$$\mathcal{L} = -\sum_{i=1}^{n} \sum_{t \in \mathcal{T}} \mathbb{1}(\tilde{\tau}_i > t) \log(S(t|x_i)) + \delta_i(1 - \mathbb{1}(\tilde{\tau}_i > t)) \log(1 - S(t|x_i)) \qquad \text{(B.4)}$$

which suffers from censoring-induced covariate shift due to the interaction of $\delta_i(1 - \mathbb{1}(\tilde{\tau}_i > t))$; i.e. only non-censored individuals contribute to the 'negative class' $\mathbb{1}(\tilde{\tau}_i \leq t)$, an effect that gets larger for $t$ large.

The multi-task logistic regression approach proposed in [14] is a variant of the more general approach described above; it uses a modeling approach based on conditional random fields [15] and jointly models all survival functions by accounting for the sequential nature of targets $\{\mathbb{1}(T > t)\}_{t \in \mathcal{T}}$ and the existance of a restricted set of 'legal' values.

### B.3 PMF-based estimation

Finally, instead of focussing on hazard or survival function, one could also estimate the PMF function; the PMF can be transformed to hazard or survival functions by realizing that $\lambda(1|x) = f(1|x)$ and $S(1|x) = (1 - f(1|x))$. This can be done by treating the survival problem as a $t_{\max}$-class classification problem with one-hot encoded labels $(\mathbb{1}(\tau_i = t))_{t \in \mathcal{T}}$ leading to the loss

$$\mathcal{L} = -\sum_{i=1}^{n} \delta_i \log(f(\tilde{\tau}_i | x_i)) \tag{B.5}$$

so that each uncensored observation contributes mainly to the estimate of $f$ at its event-time step $\tau_i$ [13] (instead of multiple time-steps as in the previous two subsections). Due to the presence of censoring indicator $\delta_i$, this suffers from censoring-induced covariate shift. As in [16]'s DeepHit, a likelihood contribution $(1 - \delta_i) \log \left( \sum_{t=\tilde{\tau}_i+1}^{t_{\max}} f(t|x_i) \right)$ marginalizing over possible outcomes for all censored observations can be added, such that they contribute to $t > \tau_i$ by signalling that their event times are larger. For correctly specified models $f$ this corresponds to optimizing the likelihood of the PMF and is hence sufficient to correct for censoring, however, otherwise this does not exactly correct for censoring-induced covariate shift.

## C   Technical details: Assumptions and Proofs

### C.1   Assumptions

In this section, we discuss and formally state the assumptions made in Section 2. As e.g. [17, 18, 19], we assume the fairly general causal structure encoded in the DAG in Figure 1. By assuming that observed data was generated from this DAG, the classical identifying assumptions (No Hidden Confounders, Censoring At Random, and Consistency) are implicitly formalized [17].

Equivalently, we can restate the assumptions using potential outcomes [20] notation. As in e.g. [21], we let $T_a$ denote the potential event time that would have been observed had treatment a been assigned, and $C = t_{\max}$ been externally set. Then, the following assumptions are implied by the DAG:

**Assumption 1** (1.a No Hidden Confounders/ Unconfoundedness). *Treatment assignment is random conditional on covariates, i.e. $T_a \perp\!\!\!\perp A | X$.*

**Assumption 2** (1.b Censoring at random). *Censoring and outcome are conditionally independent, i.e. $T_a \perp\!\!\!\perp C | X, A$.*

**Assumption 3** (1.c Consistency). *The observed outcomes are the potential outcomes under the observed intervention, i.e. if $A = a$ then $T = T_a$.*

Then, we can write

$$\begin{aligned}
\lambda^a(\tau|x) &= \mathbb{P}(T = \tau | T \geq \tau, do(A = a, C \geq \tau), X = x) \\
&= \mathbb{P}(T_a = \tau | T_a \geq \tau, do(C \geq \tau), X = x) \\
&= \mathbb{P}(T_a = \tau | T_a \geq \tau, A = a, do(C \geq \tau), X = x) \\
&= \mathbb{P}(T_a = \tau | T_a \geq \tau, C \geq \tau, A = a, X = x) \\
&= \mathbb{P}(T = \tau | T \geq \tau, C \geq \tau, A = a, X = x) \\
&= \mathbb{P}(\tilde{T} = \tau, \Delta = 1 | \tilde{T} \geq \tau, A = a, X = x) = \lambda(\tau | a, x)
\end{aligned}$$

Here, the equalities in line one and two follow by definition, line three follows by assumption 1.a, line four follows by assumption 1.b, the equality in line five follows by assumption 1.c, and the final line follows by definition.

To enable nonparametric estimation of $\lambda^a(\tau|x)$ for some fixed $\tau \in \mathcal{T}$, we additionally consider a number of conditions on the likelihood of observing certain events.

**Assumption 4** (2.a Overlap/positivity (treatment assignment)). *Treatment assignment is non-deterministic, i.e. for some $\epsilon_1 > 0$, we have that $\epsilon_1 < \mathbb{P}(A = a | X = x) < 1 - \epsilon_1$*

**Assumption 5** (2.b Positivity (censoring)). *Censoring is non-deterministic, i.e. for some $\epsilon_2 > 0$, we have that $\mathbb{P}(N_C(t) = 0 | A = a, X = x) = \mathbb{P}(C > t | A = a, X = x) => \epsilon_2$ for all $t < \tau$.*

**Assumption 6** (2.c Positivity (events)). *Not all events deterministically occur before time $\tau$, i.e. $\mathbb{P}(N_T(\tau - 1) = 0 | A = a, X = x) > \mathbb{P}(T > \tau - 1 | A = a, X = x)\epsilon_3 > 0$*

Assumptions 1.a, 1.c and 2.a are standard within the treatment effect estimation literature [2, 1]; assumptions 1.b and 2.b are standard within the literature with survival outcomes [21, 22]. Assumption 2.c is needed only if we aim to estimate $\lambda^a(t|x)$ for all $t$, otherwise it would suffice to follow a convention such as setting $\lambda^a(t|x) = 1$ whenever $\mathbb{P}(N_T(\tau - 1) = 0 | A = a, X = x) = 0$.

## C.2 Proof of proposition 1

In this section we state the proof of proposition 1 and restate two lemmas from [23] which we use within the proof.

**Notation and definitions (restated)** For fixed $a, \tau$ and representation $\Phi : \mathcal{X} \to \mathcal{R}$, let $\mathbb{P}_0^\Phi$, $\mathbb{P}_{a,\tau}^\Phi$ and $\mathbb{P}_{a,\tau}^{w,\Phi}$ denote the baseline, observational and weighted observational distribution w.r.t. the representation $\Phi$. Define the pointwise losses

$$
\begin{aligned}
l_{h,\mathbb{Q}}(x; a, \tau) &\stackrel{\text{def}}{=} \mathbb{E}_{Y(\tau)|x, a \sim \mathbb{Q}}[\ell(Y(\tau), h(\Phi(X)))|X = x, A = a] \\
l_{h,\mathbb{Q}^\Phi}(\phi; a, \tau) &\stackrel{\text{def}}{=} \mathbb{E}_{Y(\tau)|\phi, a \sim \mathbb{Q}^\Phi}[\ell(Y(\tau), h(\Phi))|\Phi = \phi, A = a]
\end{aligned}
\tag{C.1}
$$

of (hazard) hypothesis $h : \mathcal{R} \to [0, 1]$ w.r.t. distributions in covariate and representation space, respectively.

Further, define the integral probability metric distance (IPM) w.r.t. a function class $\mathcal{G}$ as

$$
\text{IPM}_{\mathcal{G}}(\mathbb{P}, \mathbb{Q}) = \sup_{g \in \mathcal{G}} \left| \int g(x)(\mathbb{P}(x) - \mathbb{Q}(x))dx \right|
\tag{C.2}
$$

Define the excess target information loss $\eta_\Phi^\ell(h)$ analogously to [23] as

$$
\eta_\Phi^\ell(h) \stackrel{\text{def}}{=} \mathbb{E}_{X \sim \mathbb{P}_0}[\xi_{\mathbb{P}^\Phi, \mathbb{P}}(X) - \xi_{\mathbb{P}_{a,\tau}^{w,\Phi}, \mathbb{P}}(X)]
\tag{C.3}
$$

with

$$
\xi_{\mathbb{Q}^\Phi, \mathbb{Q}}(x) \stackrel{\text{def}}{=} l_{h,\mathbb{Q}^\Phi}(\phi; a, \tau) - l_{h,\mathbb{Q}}(x; a, \tau)
\tag{C.4}
$$

**Preliminaries**

**Lemma 1** (Adapted from Lemma A.3 in [23]).

$$
\mathbb{E}_{X \sim \mathbb{Q}}[\ell_{h,\mathbb{Q}}(X; a, \tau)] = \mathbb{E}_{\Phi \sim \mathbb{Q}^\Phi}[\ell_{h,\mathbb{Q}^\Phi}(\Phi; a, \tau)]
$$

*Proof.* This proof is adapted to our notation and setting from [23] and stated for completeness. Let $y = y(\tau)$ and $z = \Phi(x)$

$$
\begin{aligned}
\mathbb{E}_{\Phi \sim \mathbb{Q}^\Phi}[\ell_{h,\mathbb{Q}^\Phi}(\Phi; a, \tau)] &= \int_{z,y} \mathbb{Q}^\Phi(z, y)\ell(y, h(z))dzdy \\
&= \int_{z,y} \ell(y, h(z)) \int_{x:z=\Phi(x)} \mathbb{Q}(x, y)dxdzdy \\
&= \int_{x,y} \mathbb{Q}(x, y) \int_z \mathbb{1}\{z = \Phi(x)\}\ell(y, h(z))dzdxdy \\
&= \int_{x,y} \mathbb{Q}(x, y)\ell(y, h(\Phi(x))dxdy \\
&= \mathbb{E}_{X \sim \mathbb{Q}}[\ell_{h,\mathbb{Q}}(X; a, \tau)]
\end{aligned}
$$

$\square$

**Lemma 2** (Adapted from Lemma A.4 in [23])**.**

$$\mathbb{E}_{X \sim \mathbb{P}_0}[\ell_{h,\mathbb{P}_0}(X;a,\tau)] = \mathbb{E}_{\Phi \sim \mathbb{P}_0^\Phi}[l_{h,\mathbb{P}_{a,\tau}^{w,\Phi}}(\Phi;a,\tau)] + \eta_\Phi^l(h)$$

*Proof.* This proof is adapted to our notation and setting from [23] and stated for completeness.

$$
\begin{aligned}
\mathbb{E}_{X \sim \mathbb{P}_0}[\ell_{h,\mathbb{P}_0}(X;a,\tau)] &= \mathbb{E}_{X \sim \mathbb{P}_0}[\mathbb{E}_{Y(\tau)|x,a \sim \mathbb{Q}}[\ell(Y(\tau), h(\Phi(X)))|X=x, A=a]] \\
&= \mathbb{E}_{\Phi \sim \mathbb{P}_0^\Phi}[\ell_{h,\mathbb{P}_0^\Phi}(\Phi;a,\tau)] \text{ (by Lemma 1)} \\
&= \mathbb{E}_{\Phi \sim \mathbb{P}_0^\Phi}[\ell_{h,\mathbb{P}_{a,\tau}^{w,\Phi}}(\Phi;a,\tau)] + \mathbb{E}_{\Phi \sim \mathbb{P}_0^\Phi}[\ell_{h,\mathbb{P}_0^\Phi}(\Phi;a,\tau)] - \mathbb{E}_{\Phi \sim \mathbb{P}_0^\Phi}[\ell_{h,\mathbb{P}_{a,\tau}^{w,\Phi}}(\Phi;a,\tau)] \\
&= \mathbb{E}_{\Phi \sim \mathbb{P}_0^\Phi}[\ell_{h,\mathbb{P}_{a,\tau}^{w,\Phi}}(\Phi;a,\tau)] + \mathbb{E}_{\Phi \sim \mathbb{P}_0^\Phi}[\xi_{\mathbb{P}_0^\Phi,\mathbb{P}}(X) - \xi_{\mathbb{P}_{a,\tau}^{w,\Phi},\mathbb{P}}(X)] \\
&= \mathbb{E}_{\Phi \sim \mathbb{P}_0^\Phi}[l_{h,\mathbb{P}_{a,\tau}^{w,\Phi}}(\Phi;a,\tau)] + \eta_\Phi^l(h)
\end{aligned}
$$

where the second to last line follows as $l_{h,\mathbb{P}}(x;a,\tau)$ cancels in $\eta_\Phi^l(h)$ □

**Proof of proposition 1**

**Proposition 1** (Restated)**.** *Assume there exists a constant $C_\Phi > 0$ s.t. $C_\Phi^{-1} \ell_{h,\mathbb{P}_{a,\tau}^{w,\Phi}}(\phi,a,\tau) \in \mathcal{G}$ for some family of functions $\mathcal{G}$. Then we have that*

$$\underbrace{\mathbb{E}_{X \sim \mathbb{P}_0}[\ell_{h,\mathbb{P}}(X;a,\tau)]}_{\textit{Target Risk}} \le \underbrace{\mathbb{E}_{X \sim \mathbb{P}_{a,\tau}}[w_{a,\tau}(X)\ell_{h,\mathbb{P}}(X;a,\tau)]}_{\textit{Weighted observational risk}} + C_\Phi \underbrace{\textit{IPM}_G(\mathbb{P}_0^\Phi, \mathbb{P}_{a,\tau}^{w,\Phi})}_{\textit{Distance in } \Phi\textit{-space}} + \underbrace{\eta_\Phi^l(h)}_{\textit{Info loss}} \quad \text{(C.5)}$$

*Proof.* By Lemma 2,

$$\mathbb{E}_{X \sim \mathbb{P}_0}[\ell_{h,\mathbb{P}_0}(X;a,\tau)] = \mathbb{E}_{\Phi \sim \mathbb{P}_0^\Phi}[l_{h,\mathbb{P}_{a,\tau}^{w,\Phi}}(\Phi;a,\tau)] + \eta_\Phi^l(h)$$

Further,

$$
\begin{aligned}
\mathbb{E}_{\Phi \sim \mathbb{P}_0^\Phi}[l_{h,\mathbb{P}_{a,\tau}^{w,\Phi}}(\Phi;a,\tau)] - \mathbb{E}_{\Phi \sim \mathbb{P}_{a,\tau}^{w,\Phi}}[l_{h,\mathbb{P}_{a,\tau}^{w,\Phi}}(\Phi;a,\tau)] &= \int_\phi \ell_{h,\mathbb{P}_{a,\tau}^{w,\Phi}}(\phi;a,\tau)(\mathbb{P}_0^\Phi(\phi) - \mathbb{P}_{a,\tau}^{w,\Phi}(\phi))d\phi \\
&= C_\Phi \int_\phi \frac{\ell_{h,\mathbb{P}_{a,\tau}^{w,\Phi}}(\phi;a,\tau)}{C_\Phi}(\mathbb{P}_0^\Phi(\phi) - \mathbb{P}_{a,\tau}^{w,\Phi}(\phi))d\phi \\
&\le C_\Phi \sup_{g \in \mathcal{G}} \left| \int_\phi g(\phi)(\mathbb{P}_0^\Phi(\phi) - \mathbb{P}_{a,\tau}^{w,\Phi}(\phi))d\phi \right| \\
&= C_\Phi \text{IPM}_G(\mathbb{P}_0^\Phi, \mathbb{P}_{a,\tau}^{w,\Phi})
\end{aligned}
$$

Thus

$$
\begin{aligned}
\mathbb{E}_{X \sim \mathbb{P}_0}[\ell_{h,\mathbb{P}_0}(X;a,\tau)] &\le \mathbb{E}_{\Phi \sim \mathbb{P}_{a,\tau}^{w,\Phi}}[l_{h,\mathbb{P}_{a,\tau}^{w,\Phi}}(\Phi;a,\tau)] + C_\Phi \text{IPM}_G(\mathbb{P}_0^\Phi, \mathbb{P}_{a,\tau}^{w,\Phi}) + \eta_\Phi^l(h) \\
&= \mathbb{E}_{X \sim \mathbb{P}_{a,\tau}^w}[\ell_{h,\mathbb{P}}(X;a,\tau)] + C_\Phi \text{IPM}_G(\mathbb{P}_0^\Phi, \mathbb{P}_{a,\tau}^{w,\Phi}) + \eta_\Phi^l(h)
\end{aligned}
$$

where the last line follows by Lemma 1 and the unconfoundedness and censoring at random assumptions, by which $\ell_{h,\mathbb{P}}(X;a,\tau) = \ell_{h,\mathbb{P}_{a,\tau}^w}(X;a,\tau)$

□

## D  Implementation

We discuss implementation of SurvITE and baselines in turn below. The source code for SurvITE is available in `https://github.com/chl8856/survITE`. Throughout the experiments, training SurvITE and its variants takes approximately 30 minutes to 1 hour on a single GPU machine[2].

---

[2]The specification of the machine is: CPU – Intel Core i7-8700K, GPU – NVIDIA GeForce GTX 1080Ti,and RAM – 64GB DDR4.

## D.1 SurvITE

Throughout the experiments, we implement SurvITE utilizing 3-layer fully-connected network (FC-Net) with 100 nodes in each layer for the representation estimator $\Phi$, and 2-layer FC-Net with 100 nodes in each layer for each hypothesis estimator $h_{a,t}$, respectively. The parameters $(\theta_\Phi, \theta_h)$ are initialized by Xavier initialization [24] and optimized via Adam optimizer [25] with learning rate of 0.001 and dropout probability of 0.3. We choose the balancing coefficient $\beta$ within a set of possible candidates $\mathcal{B} = \{1., 0.1, 0.01, 0.001, 0.0001\}$ utilizing a grid search. More specifically, we select the highest value in $\mathcal{B}$ that does not compromise its discriminative performance (i.e. C-Index) based on the validation set (i.e., 20% of the training set) to guarantee that the learned representation is balanced as much as possible to adjust for the covariate shift while being informative about the survival predictions. The effect of the balancing coefficient is further investigated in Section F.2.

**Finite-Sample Wasserstein Distance.** For the finite sample approximation of the Wasserstein distance, we use Algorithm 1 with the entropic regularization strength set to $\lambda = 10$ and the number of Sinkhorn iterations set to $S = 10$ following the implementation in [26, 27]. Thus, given two sets of samples $\mathcal{B}_0, \mathcal{B}_1$ based on the treatment-group time-step combinations, we can compute $Wass\big(\{\Phi(\mathbf{x}_i)\}_{i \in \mathcal{B}_0}, \{\Phi(\mathbf{x}_i)\}_{i \in \mathcal{B}_1}\big)$ based on Algorithm 1.

---

**Algorithm 1** Pseudo-code for Finite Sample Wasserstein Distance

---

**Input:** Set $\mathcal{B}_0$, $\mathcal{B}_1$, entorpic regularization parameter $\lambda \in \mathbb{R}$, the number of Sinkhorn iterations $S$, representation $\theta_\Phi$
$n_1 = |\mathcal{B}_1|$ and $n_0 = |\mathcal{B}_0|$
$a = \frac{1}{n_1}\mathbf{1} \in \mathbb{R}^{n_1}$ and $b = \frac{1}{n_0}\mathbf{1} \in \mathbb{R}^{n_0}$
$M^{(i,j)} = \|\Phi(\mathbf{x}_i) - \Phi(\mathbf{x}_j)\|_2 \ \forall i \in \mathcal{B}_1, \forall j \in \mathcal{B}_0$
$K = \exp(-\lambda M)$
$\tilde{K} = \text{diag}(a^{-1})K$
Initialize $u = a$
**for** $s = 1, \cdots, S$ **do**
    $u = 1./(\tilde{K}(b./(K^T u)))$
**end for**
$v = b./(K^T u).$
$T_\lambda^* = \text{diag}(u)K\text{diag}(v)$
**Output:** $Wass\big(\{\Phi(\mathbf{x}_i)\}_{i \in \mathcal{B}_0}, \{\Phi(\mathbf{x}_i)\}_{i \in \mathcal{B}_1}\big) \approx \sum_{i,j} T_\lambda^{*(i,j)} M^{(i,j)}$

---

**Smoothing and Parameter Sharing.** Employing a separate FC-Net at each time step provides sufficient capacity to estimate the hazard function accurately. However, this can be computationally burdensome as the number of parameters linearly increases with the number time steps considered in the study, and may result in having hypothesis estimators overfitted at the later time steps due to the scarcity of samples at those time steps. To avoid such issues, one can employ coarser time intervals for discritization or non-uniform time intervals (as in our experiments on the Twins dataset) such that finer time intervals are used in the earlier time steps and coarser time intervals are used in the later time steps to guarantee a sufficient amount of samples for training each hypothesis network. In addition to these immediate solutions, one could consider two different remedies that slightly change our model design:

- **Smoothing regularization**: We introduce an auxiliary regularization term that smooths the hazard estimators across time steps for each treatment group. This encourages the hazard estimators not to deviated too much from those at adjacent time steps. Formally, the smoothing regularization is given by

$$\mathcal{L}_{smoothing}(\theta_h) = \sum_{a \in \{0,1\}} \sum_{t=1}^{t_{\max}} \|\theta_{h_{a,t}} - \theta_{h_{a,t-1}}\|_2^2.$$

- **Parameter Sharing**: Instead of employing a separate FC-Net for each time step, we implement a single FC-Net for each treatment group that is shared throughout the time steps i.e., $t \in \mathcal{T}$, taking both the representation $\Phi(x)$ and $t$ as input to the network. Formally, the hazard function is defined as $h_a : \mathcal{R} \times \mathcal{T} \to [0, 1]$.

We present experimental results using these approaches in Section F.3.

## D.2 Details of Baselines

We compared SurvITE with baselines ranging from commonly used survival methods to the state-of-the-art HTE methods based on deep neural networks. The details of how we implemented the benchmarks are described as the following:

- **Cox**[3] [29], **RSF**[3] [30], and **DeepHit**[4]: When there are treatments, we use these models in a two-model (T-learner) approach by training a separate model using samples in the treated ($A = 1$) and controlled ($A = 0$) groups, respectively. For Cox, we set the coefficient for ridge regression penalty as $\alpha = 0.001$. For RSF, we use the default hyper-parameter setting (i.e., *n_estimators* $= 100$ using a survival tree as the baseline estimator and *min_samples_leaf* $= 3$ without maximum depth restriction). For DeepHit, we use utilize the 3-layer FC-Net with 100 nodes in each layer. We choose the DeepHit's hyper-parameters $\alpha, \sigma$ from a set of possible candidates $\{0.001, 0.01, 0.1, 1, 10\}$ and $\{0.01, 0.1, 1.10\}$, respectively.

- **LR-sep**: We utilize the long data format as described in Section 2 of the manuscript and train a separate logistic regression model[5] at each time step $t \in \mathcal{T}$ to solve the hazard classification problem utilizing only "at-risk" samples whose time-to-event/censoring is at or after $t$. Formally, the logistic regression models are trained based on the log-loss in (B.3). When there are treatments, we use LR-sep in a two-model (T-learner) approach by training a separate model using samples in the treated ($A = 1$) and controlled ($A = 0$) groups, respectively.

- **CSA**[6] [31]: We use the CSA-INFO model of [31], where we use its generative capabilities to approximate target quantities via monte-carlo sampling. We use the code and specifications provided by the authors, in particular we use a hidden dimension of 100, set the imbalance penalty $\alpha = 100$ and train for 300 epochs. To create monte carlo approximations, we sample 1000 times from the model for each observation in the test set.

- **SurvITE (CFR-1)** and **SurvITE (CFR-2)**: We consider two variants of SurvITE by replacing our $\mathcal{L}_{ipm}(\theta_\phi)$ with a balancing term based on the CFRNet[7] proposed in [1]:

  - **SurvITE (CFR-1)** creates a representation balancing treatment groups at baseline only which is formally given as:

  $$\mathcal{L}_{ipm}(\theta_\phi) = Wass\big(\{\Phi(x_i)\}_{i:a_i=1}, \{\Phi(x_i)\}_{i:a_i=0}\big) \qquad (D.1)$$

  - **SurvITE (CFR-2)** creates a representation optimizing for balance of treatment groups *at each time step*

  $$\mathcal{L}_{ipm}(\theta_\phi) = \sum_{t=1}^{t_{\max}} Wass\big(\{\Phi(x_i)\}_{i:\tilde{\tau}_i \geq t, a_i=1}, \{\Phi(x_i)\}_{i:\tilde{\tau}_i \geq t, a_i=0}\big) \qquad (D.2)$$

  Note that, in both variants, there is no balancing towards $\mathbb{P}_0$. We implement SurvITE (CFR-1) and SurvITE (CFR-2) with the same network architecture and hyper-parameters with those of SurvITE.

# E Dataset Description and Experimental Setup

## E.1 Synthetic Experiments

In this section, we present some illustrations of the properties of the synthetic DGPs. Recall that $\lambda_w(\tau|x)$ is the same across all settings, therefore we focus here on S3 to analyze the interplay of selection bias and event processes. In Fig. E.1, we present histograms of event times in S3 for different degrees of selection bias. Note that there is a positive treatment effect (treatment reduces

---

[3]Python package `scikit-survival` [28]
[4]`https://github.com/chl8856/DeepHit`
[5]Python package `scikit-learn`
[6]`https://github.com/paidamoyo/counterfactual_survival_analysis`
[7]`https://github.com/clinicalml/cfrnet`

event probabilities) encoded in our DGP; this is clearly visible in the left panel without selection bias as consistently more events occur for control than for treated group. As we add selection bias, it seems that treatment has a *negative effect* on survival after time 10, as more events occur in the treatment group. This correlation is spurious: as $X_2$ linearly increases event hazard, and treatment is selected based on $X_2$ (rightmost panel) or based on a variable correlated with it (middle panel) it seems *as if* treatment increases mortality. Note that this is not the case for $t < 10$ because $X_1$ enters $\lambda_w(\tau|x)$ in squared form.

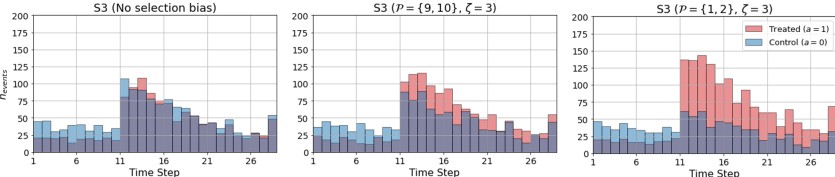

Figure E.1: Histograms of event times in S3 for different degrees of selection bias; no selection (left), no overlapping selection covariates (middle) and full overlap (right)

In Figure E.2 we further analyze the interplay between event-induced covariate shift and selection bias by considering the distribution of $X_1$ in the at-risk population over time. As $-X_1^2$ appears in the hazard, individuals with small magnitude of $X_1$ have lower probability of survival – this becomes visible for $\zeta = 0$ as the at-risk histogram flattens out over time. Because $X_1$ enters $e(x)$ linearly, when we add selection bias ($\zeta > 0$), we observe that the populations not only differ already at baseline, but that the difference appears to become more extreme over time – this is precisely because the overlapping parts of the population ($|X_1|$ small) have larger event probability, so that the event-induced shift further amplifies the selection boas over time.

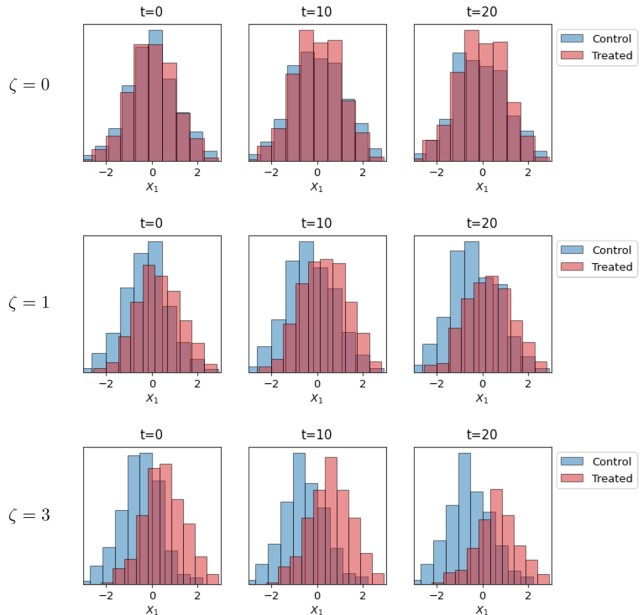

Figure E.2: Histograms of $X_1$ in the at-risk population by time (left to right) and $\zeta$ (top to bottom) for S3 with $\mathcal{P} = \{1, 2\}$, highlighting covariate shift due to selection bias (across $\zeta$) and occurred events (across $t$)

## E.2 Semi-Synthetic Experiments: Twins

This dataset is derived from all births in the USA between 1989–1991 [32] where we only focus on the twins. We artificially create a binary treatment such that $a = 1 (a = 0)$ denotes being born the heavier (lighter). The outcome of interest is the time-to-mortality (in days) of each of the twins in their first year, thus administratively censored at $t = 365$. Since we have records for both twins, we treat their

time-to-event outcomes as two potential outcomes, i.e., $\tilde{\tau}(1)$ and $\tilde{\tau}(0)$, with respect to the treatment assignment of being born heavier. As previously used in [33, 34], we obtained 30 features (which has 39 feature dimension after one-hot encoding on categorical features) for each twin relating to the parents, the pregnancy, and the birth (e.g., marital status, race, residence, number of previous births, pregnancy risk factors, quality of care during pregnancy, and number of gestation weeks prior to birth). We only chose twins weighing less than 2kg and without missing features. To create an observational time-to-event dataset, we selectively observed one of the two twins **(no censoring)** with selection bias and **(censoring)** with both selection bias and censoring bias as follows: the treatment assignment is given by $a|x \sim \texttt{Bern}(\sigma(w_1^\top x + e))$ where $w \sim \texttt{Uniform}(-0.1, 0.1)^{39 \times 1})$ and $e \sim \mathcal{N}(0, 1^2)$, and the time-to-censoring is given by $C \sim \texttt{Exp}(100 \cdot \sigma(w_2^\top x))$ where $w_2 \sim \mathcal{N}(0, 1^2)$.

For continuous-time models (i.e., Cox, RSF, and CSA), we use the original time resolution in days without discarding any information. For discrete-time models (i.e., LR-sep, SurvITE , and variants of SurvITE), we use a non-uniform discretization – i.e. resolution of days in the first 30 days and months after the first 30 days – because most of the events are concentrated in the first 30 days (approximately 87% of the events occur within that period).

### E.3 Performance Metrics

Once SurvITE (or SurvIHE) is trained, we can simply estimate the (treatment-specific) survival function based on the estimated hazard functions as the following:

$$\hat{S}^a(\tau|x) = \prod_{t \leq \tau} \left(1 - h_{a,t}(\Phi(x))\right) \qquad \text{for } a \in \{0, 1\}. \tag{E.1}$$

**Heterogeneous Treatment Effects.** For synthetic experiments where we have the ground-truth treatment-specific survival functions i.e., $S^1(\tau|x)$ and $S^0(\tau|x)$, we evaluate $HTE_{surv}(\tau|x) = S^1(\tau|x) - S^0(\tau|x)$ and $HTE_{rmst}(x; L) = \sum_{t_k \leq L} \left(S^1(t_k|x) - S^0(t_k|x)\right) \cdot (t_k - t_{k-1})$ in terms of the averaged root mean squared error (RMSE) of the estimation:

$$\epsilon_{HTE_{surv}}(t) = \sqrt{\frac{1}{n} \sum_{i=1}^n \left(HTE_{surv}(t|x_i) - \widehat{HTE}_{surv}(t|x_i)\right)^2}, \tag{E.2}$$

$$\epsilon_{HTE_{rmst}}(L) = \sqrt{\frac{1}{n} \sum_{i=1}^n \left(HTE_{rmst}(x_i; L) - \widehat{HTE}_{rmst}(x_i; L)\right)^2}. \tag{E.3}$$

Here $\widehat{HTE}_{surv}(t|x) = \hat{S}^1(\tau|x) - \hat{S}^0(\tau|x)$ and $\widehat{HTE}_{rmst}(x; L) = \sum_{t_k \leq L} \left(\hat{S}^1(t_k|x) - \hat{S}^0(t_k|x)\right) \cdot (t_k - t_{k-1})$ where $(t_k - t_{k-1})$ may vary depending on how the continuous time is discretized (e.g., non-uniform time intervals for the Twins dataset).

For semi-synthetic experiments where we have the ground-truth treatment-specific time-to-event outcomes but not the treatment-specific survival functions, we only report $\epsilon_{HTE_{rmst}}(L)$ in (E.3) where the ground-truth $HTE_{rmst}(x; L)$ is defined in terms of the ground-truth time-to-event outcomes, i.e., $HTE_{rmst}(x; L) = (\min(T(1), L) - \min(T(0), L))$ where $T(1)$ and $T(0)$ are the time-to-event given $a = 1$ and $a = 0$, respectively.

**(Treatment-Specific) Survival Functions.** For evaluating the estimation performance of the (treatment-specific) survival functions, we evaluate the averaged RMSE of these estimations as the following:

$$\epsilon_{S^a}(t) = \sqrt{\frac{1}{n} \sum_{i=1}^n \left(S^a(t|x_i) - \hat{S}^a(t|x_i)\right)^2}. \tag{E.4}$$

**Discriminative Performance.** For assessing the survival predictions of all the survival models with respect to how well the predictions discriminate among individual risks, we use the concordance index (C-Index) [35]:

$$\text{C-Index}(t) = \mathbb{P}\left(\hat{S}(t|x_i) < \hat{S}(t|x_j) \big| \tilde{\tau}_i < \tilde{\tau}_j, \tilde{\tau}_i \leq t, \delta_i = 1\right) \tag{E.5}$$

where $\hat{S}(t|x) = a \cdot \hat{S}^1(t|x) + (1-a) \cdot \hat{S}^0(t|x)$ is the survival prediction given treatment $a$. The resulting C-Index in (E.4) tells us how well the given survival model discriminates the individual risks among the events that occur before or at time $t$.

# F    Additional Experiments

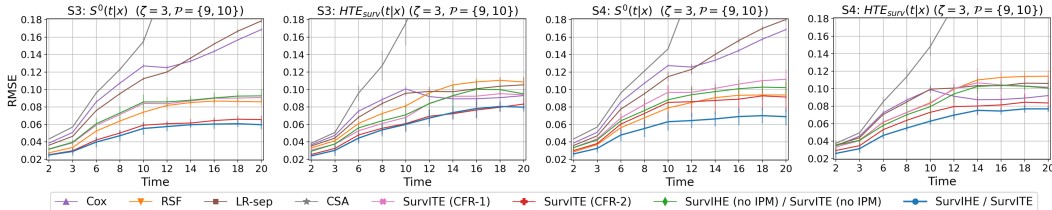

Figure F.1: RMSE of estimating the treatment-specific survival function $S^0(t|x)$ and the treatment effect $HTE_{surv}(t|x)$ for different time steps across synthetic settings (Lower is better). Averaged across 5 runs; the error bar indicates 95%-confidence interval.

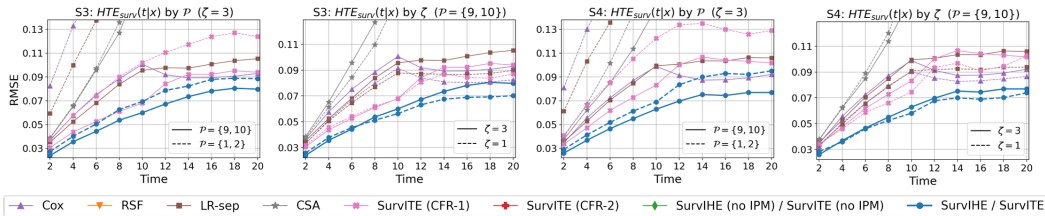

Figure F.2: RMSE of estimating the treatment effect $HTE_{surv}(t|x)$ for different time steps across synthetic settings (Lower is better) for methods not presented in the main text. Averaged across 5 runs.

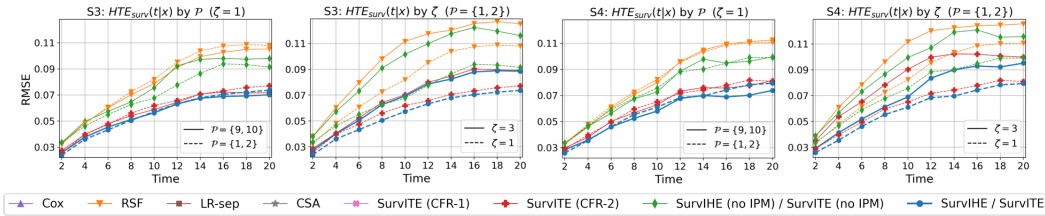

Figure F.3: RMSE of estimating the treatment effect $HTE_{surv}(t|x)$ for different time steps across synthetic settings (Lower is better) for settings not presented in the main text. Averaged across 5 runs.

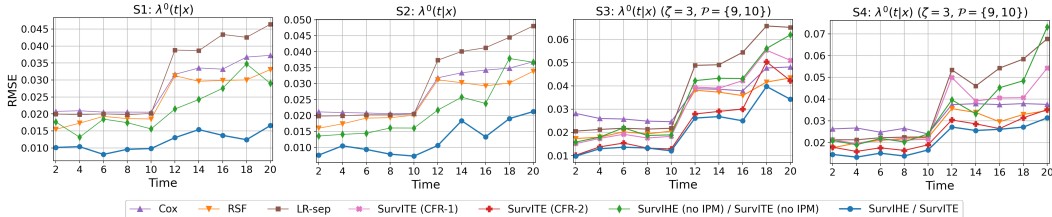

Figure F.4: RMSE of estimating the treatment-specific hazard function $\lambda^0(t|x)$ for different time steps across synthetic settings (Lower is better). Averaged across 5 runs.

## F.1    Additional Results on the Synthetic Experiments

In this subsection, we report the additional results on the synthetic experiments that were not provided in the manuscript due space constraints.

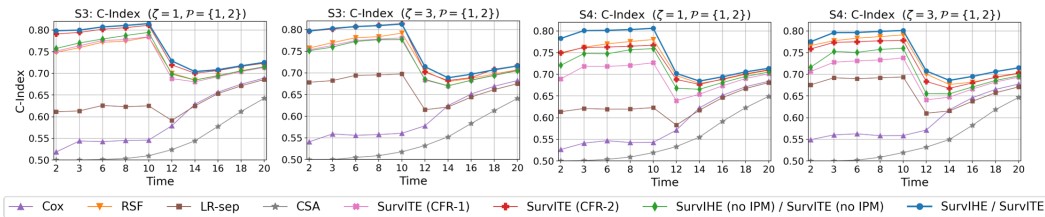

Figure F.5: C-Index for different time steps for S3 and S4 with $\zeta = 3$ and $\mathcal{P} = \{1, 2\}$ (Higher is better). Averaged across 5 runs.

Table F.1: RMSE on estimation of $S^0(t|x)$ and $\text{HTE}_{surv}(x)$ (mean $\pm$ 95%-CI) for the synthetic settings S3 and S4 with $\zeta = 3$ and $\mathcal{P} = \{9, 10\}$ at $L = 10$.

| Methods | RMSE on $S^0(t|x)$ | | RMSE on $HTE_{surv}(t|x)$ | |
| :---: | :---: | :---: | :---: | :---: |
| | S3 | S4 | S3 | S4 |
| Cox | 0.127±0.002 | 0.127±0.001 | 0.101±0.004 | 0.099±0.004 |
| RSF | 0.074±0.005 | 0.079±0.005 | 0.081±0.003 | 0.084±0.004 |
| LR-sep | 0.112±0.002 | 0.115±0.006 | 0.096±0.003 | 0.099±0.005 |
| DeepHit | 0.095±0.012 | 0.087±0.003 | 0.107±0.014 | 0.095±0.007 |
| CSA | 0.155±0.005 | 0.147±0.001 | 0.176±0.025 | 0.148±0.011 |
| SurvITE (no IPM) | 0.086±0.008 | 0.088±0.011 | 0.071±0.011 | 0.079±0.012 |
| SurvITE (CFR-1) | 0.084±0.003 | 0.097±0.009 | 0.068±0.009 | 0.083±0.005 |
| SurvITE (CFR-2) | 0.059±0.003 | 0.085±0.020 | 0.061±0.009 | 0.073±0.011 |
| SurvITE | **0.055±0.007** | **0.063±0.010** | **0.060±0.009** | **0.063±0.004** |

In Table F.1, we report the performance comparison using DeepHit with respect to the estimations on both $S^0(t|x)$ and $HTE_{surv}(t|x)$ for the synthetic settings S3 and S4 with $\zeta = 3$ and $\mathcal{P} = \{9, 10\}$ at $L = 10$. We observe that DeepHit performs worse than the SurvITE architecture without IPM term, indicating that our model architecture alone is better suited for estimation of treatment-specific survival functions (note that [16] focused mainly on discriminative (predictive) performance, and not on the estimation of the survival function itself). Therefore, upon addition of the IPM-terms, the performance gap between SurvITE and DeepHit only becomes larger.

Figure F.1 shows the performance of estimations on $S^0(t|x)$ and $HTE_{surv}(t|x)$ with error bars (omitted in the main text for readability), Figure F.2 shows the performance of $HTE_{surv}(t|x)$ estimation for survival methods that were not presented in the main text to ensure readability, and Figure F.3 shows the performance of $HTE_{surv}(t|x)$ estimation for synthetic scenarios (combinations of $\mathcal{P}$ and $\zeta$) not provided in the main text due to space constraints. In all cases, we observe that SurvITE (/SurvIHE) outperforms all other methods.

In Figure F.4 we present the RMSE of estimating the *hazard* function instead of the *survival* function as in the main text[8]. The results for hazard estimation largely mimic the ones presented in the main text; in particular, we observe that SurvITE (/SurvIHE) performs best throughout. Notably, the gaps in performance across all methods at later time-steps appear somewhat smaller for hazard than for survival functions; this is expected as the errors on hazards accumulate when the survival function is computed from them.

In addition, in Figure F.5, we report the discriminative performance of the various survival models for synthetic scenarios S3 and S4 with $\zeta = 3$ and $\mathcal{P} = \{1, 2\}$ across different time steps. We evaluate the discriminative performance in terms of the C-index defined in (E.4). The figure shows that SurvITE performs the best also in terms of discriminative performance throughout different scenarios and different time steps due to the accurate estimation of the treatment-specific survival functions.

---

[8]Note that we excluded CSA-INFO from this plot, as it only outputs real-valued time-to-event predictions, which makes it unclear how to best use it to directly estimate hazard functions.

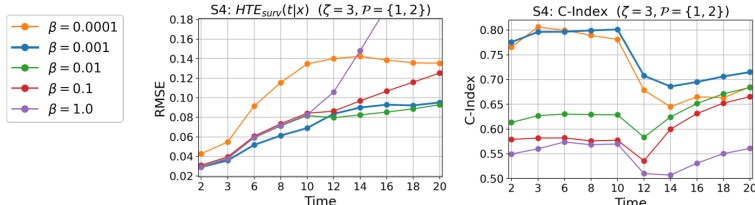

Figure F.6: RMSE of estimating the treatment effect $HTE_{surv}(t|x)$ and C-index of the survival predictions with different $\beta$ on S4 with $\zeta = 3$ and $\mathcal{P} = \{1, 2\}$. Averaged across 5 runs.

## F.2 Sensitivity Analysis

In this subsection, we investigate the effect of the balancing coefficient $\beta$ in Figure F.6 on the estimation performance of the HTE, and the discriminative performance of the survival predictions. As expected, Figure F.6 shows that SurvITE with a proper amount of IPM regularization improves the treatment effect estimation: imposing too much regularization will make the representation estimator unable to maintain important information for estimating treatment-specific hazard functions while setting regularization too low will not balance the representation from the different sources of covariate shift. Similarly, if the representation is balanced too much, it will lose discriminative power which will eventually make the trained model less useful. In this context, due to the absence of counterfactual information in practice, we propose to select the balancing coefficient $\beta$ by increasing the value starting from the lowest value in the set of possible candidates $\{1., 0.1, 0.01, 0.001, 0.0001\}$ as long as the method maintains good discriminative performance on the validation set (and stop when discriminative performance deteriorates). In our experiments, we choose $\beta = 0.001$, which is the largest value that provides good discriminative performance based on the validation set (see the right hand panel in Figure F.6).

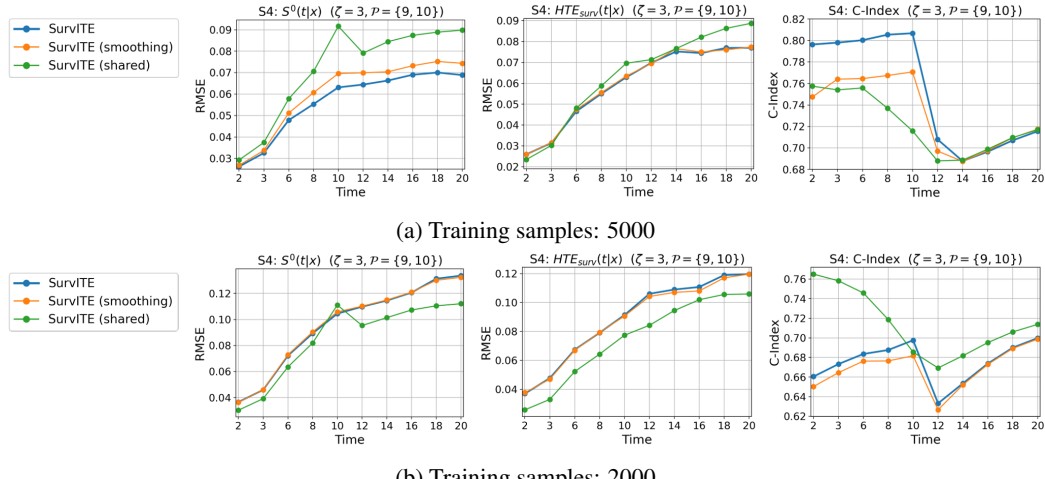

(a) Training samples: 5000

(b) Training samples: 2000

Figure F.7: RMSE of estimating the treatment-specific survival function $S^0(t|x)$, that of the treatment effect $HTE_{surv}(t|x)$, and C-index of the survival predictions $S(t|x)$ using smoothing and parameter sharing on S4 with $\zeta = 3$ and $\mathcal{P} = \{9, 10\}$. Averaged across 5 runs.

## F.3 SurvITE Variants with Smoothing and Sharing Parameters

In this subsection, we further investigate SurvITE variants with techniques that can address the practical issue of potentially having too many separate hypothesis estimators for large $t_{max}$. Figure F.7 compares the estimation performance of the treatment-specific survival functions, estimation performance of the HTE, and the discriminative performance of survival predictions. When the models are trained with a sufficient number of samples (here: 5000 training samples), the smoothing regularization maintains a very similar performance in terms of estimating the treatment-specific

survival functions and the HTE while sacrificing its discriminative performance at early time step. Sharing the parameters of hypothesis estimator across different time steps suffers more performance loss (nonetheless, it still provides reasonable performance) as the flexibility of the network is more restricted. On the other hand, when the models are trained with a smaller number of samples (here: 2000 training samples), the smoothing and sharing the network parameters play significant role in improving the estimation performance.