# OpenReview forum: "SurvITE: Learning Heterogeneous Treatment Effects from Time-to-Event Data"
_NeurIPS.cc/2021/Conference — NeurIPS 2021 Poster_

### Official Review · Reviewer_2JWN · 2021-07-07

**Rating:** 7
**Confidence:** 3

**Summary:**

This paper tackles the problem of individual treatment effect prediction in survival analysis situations, which is the case the outcome variable is the time to event. In this situation, there are two additional bias(covariate shift) problems compared with the standard ITE prediction. They showed the generalization bound for this problem, then applied counter factual regression to minimize it through the IPM term.

**Limitations And Societal Impact:**

They adequately addressed the limitation in their paper.

**Main Review:**

I think the ITE prediction for the survival outcome is an interesting topic. Such a situation occurs not only in clinical studies but also in marketing, retail, and some other business situations.

The main contribution of this work is defining the generalization bound for this problem then minimize it through the CFR scheme. I think their approach technically sounds.

It was quite time-consuming for me to understand this paper. Mainly because of the lack of knowledge about the survival analysis. So I appreciate it if you could add some explanation for the problem this paper tries to solve without using the term for the survival analysis.

For minor issues

- line 156 there are two dots before "Here,"
- line 361 RMST -> RMSE?
- Is there any reason to use ITE and HTE alternately?

**Time Spent Reviewing:**

6

---

> ### Author Response · Authors · 2021-08-10
> **General introduction to the problem**
>
> Thank you for your thoughtful comments and suggestions. We are grateful for your appreciation of our problem setting and our approach! Below, we respond to your comments in turn.
>
> [1. General introduction to the problem]
>
> We indeed acknowledge that, by considering a problem setting at the intersection of two areas of research with a rich history (survival analysis and treatment effect estimation) we had to cover significant grounds in a very limited amount of space, potentially making it difficult to follow for someone less well-versed in either one area. We do note, however, that this focus on concepts arising specifically in the survival context was necessary, as it was precisely our goal to study the unique characteristics arising at this intersection of problems. Therefore, as per your suggestion, we will use the additional space in the revised manuscript to include a ‘preliminaries’ section, giving some more intuition on the setting of survival analysis and its conceptual differences with treatment effect estimation in the standard static setting.
>
> [2. Minor comments]
>
> We thank the reviewer for the minor comments. Here are our responses:
> -	RMST in line 361 stands for Restricted mean survival time
> -	Throughout, we chose to rely on the word HTE, as the use of the term ITE is somewhat disputed within the community. We use ITE only in the name of our method (SurvITE), in line with the community’s preference for catchy names.

---

> ### Author Response · Authors · 2021-08-18
> **Dear Reviewer 2JWN**
>
> Once again, we would like to thank you for your invaluable feedback! We were wondering whether our response from August 10 has sufficiently addressed your concerns. If you have any leftover comments or concerns, please let us know - we would be happy to do our utmost to address them!

---

> ### Author Response · Authors · 2021-08-23
> **Did our response address your concerns?**
>
> Dear Reviewer 2JWN,
>
> We are sincerely grateful for your time and energy in the review process!
>
> Given the limited time left in this response phase, we wanted to make a final enquiry whether our response has addressed your concerns. We would also be eager to follow up on any additional comments you may have.
>
> Thank you!
>
> Paper4461 Authors

---

### Official Review · Reviewer_4Jiu · 2021-07-16

**Rating:** 6
**Confidence:** 4

**Summary:**

In this work the authors quickly remind the reader of the pitfalls one encounter in the survival analysis framework, in order to motivate a new loss and corresponding deep model for modelling of the group-specific hazard functions.
The main contribution of this paper resides in Proposition 1: an upper bound of the population risk (unknown) in terms of quantities that can be estimated from the data. From this proposition the authors propose to minimize the upper-bound and then provide an example to motivate their intuition.

**Limitations And Societal Impact:**

Yes, the authors are aware of the potential limitations of their work in terms of fairness. There is, however, not much they can do if they want to obtain bounds.

**Main Review:**

I find this paper interesting and fairly well written (given the constraints of the NeurIPS format). The bound in Proposition 1 is not only interesting from a purely theoretical point of view but also enables practical methods, of which the authors give an example. The bound is similar to previous bounds in the literature but does not rely on invertibility, which makes it much more practical as most models in use today are not invertible and when they are (e.g. Normalizing Flows), are computationally expensive.

The authors correctly point out that it's possible to solve the same problem by reweighting the ERM loss. It would have been worthwhile to study the link between their proposed method and propensity methods as it is known that reweighting and optimal transport plans (present here through the Wasserstein metric) are equivalent and since bounds on the excess risk in the reweighted ERM framework under censorship are known (see e.g. [1], or [2, 3]).

[1] G. Ausset, S. Clémençon, and F. Portier, “Empirical Risk Minimization under Random Censorship: Theory and Practice,” 2019, Available: http://arxiv.org/abs/1906.01908
[2] W. Stute, “The Central Limit Theorem Under Random Censorship,” Ann. Statist., vol. 23, no. 2, pp. 422–439, Apr. 1995, doi: 10.1214/aos/1176324528.
[3] T. A. Gerds, J. Beyersmann, L. Starkopf, S. Frank, J. V. D. Laan, and M. Schumacher, “The Kaplan-Meier Integral in the Presence of Covariates: A Review,” From Statistics to Mathematical Finance, pp. 25–42, 2017, doi: 10.1007/978-3-319-50986-0.


I also find the choice of competing methods in the experimental section to be puzzling and in need of better motivation: the authors exclude a wide range of methods (namely the "deep" Cox extensions like [4] or [5]) that are known to often be state of the art, for reasons I do not fully understand. This is especially puzzling since they later use the vanilla Cox model as a baseline.

[4] J. Katzman, U. Shaham, J. Bates, A. Cloninger, T. Jiang, and Y. Kluger, “DeepSurv: Personalized Treatment Recommender System Using A Cox Proportional Hazards Deep Neural Network,” BMC Med Res Methodol, vol. 18, no. 1, p. 24, Dec. 2018, doi: 10.1186/s12874-018-0482-1.
[5] C. Lee, W. Zame, J. Yoon, and M. V. D. Schaar, “DeepHit: A Deep Learning Approach to Survival Analysis With Competing Risks,” 2018.


Despite those shortcomings I think, the paper is still overall a good fit for NeurIPS.

**Time Spent Reviewing:**

4

---

> ### Author Response · Authors · 2021-08-10
> **Link to reweighting methods and theory, and additional baselines**
>
> Thank you for your thoughtful comments and suggestions. We are grateful for your appreciation of our writing, theory, and method. Below, we discuss responses to your two main comments.
>
> [1. Link to reweighting methods and theory]
>
> First, we would like to point out that our general expected risk bound in Proposition 1 actually includes a placeholder for an arbitrary weighting function, meaning that the IPM/representation balancing-based approach we take using SurvITE is in principle complementary to an importance weighted approach. Indeed, as we discuss in the paragraph line 278-288, we actually tested augmenting SurvITE’s loss function with plug-in estimates of the true importance weight (which, as discussed in line 199-208, depends on propensity score and the survival curves of event and censoring processes) but did not find this to improve its performance. It could be an interesting avenue for future methodological work to test whether replacing the plug-in approach with weights that are learned end-to-end as in [10] or [12] could lead to better results.
>
> Second, we agree that it would be an interesting next step for future theoretical work to relate the expected target risk not only to the expected reweighted observational risk as in Proposition 1, but also to its empirical analogue to study convergence (which we do not do in this paper). The techniques used in Ausset et al (2019) or [10, 32] would appear to be a good starting point for doing so. Nonetheless, we would also like to point out that the linked references, while indeed considering ERM under censorship, appear to focus on slightly different target parameters than our paper: Ausset et al (2019) appear to target the risk of the regression function $E[T|X=x]$, and Gerds et al (2017) consider expected risks that can be written as $\int_x \int_0^\infty l(t, x) F(dt|x)H(dx)$, while we target the hazard function which requires integration w.r.t. $F(dt|t>\tau, x)$.
>
> [2. Additional Baselines: DeepHit]
>
> We would like to begin by re-emphasizing that our main goal in this paper was to study the unique characteristics of the problem of inferring heterogeneous treatment effects on survival. Through a thorough theoretical treatment, we found that a number of covariate shifts from multiple sources arise due to the inherent  complexities of the time-to-event setting. As a consequence, our main interest in the experimental evaluation lay in the question of whether correct handling of said covariate shifts would improve upon naïve solutions to the problem.
>
> Therefore, we considered the most relevant baselines to be ablations of our own model that incorrectly handle the shifts: by fixing the model architecture and varying only how shifts are handled, we were able to isolate the effect of balancing terms (target effect) from the effect of ‘the architecture’ (nuisance effect). This is why we included three additional variants of our model:
>
> -	SurvITE (no IPM): only the architecture, does not handle shifts
>
> -	SurvITE (CFR-1): architecture with one IPM term that creates a representation that balances treatment groups at baseline only (naïve extension of [9] to time-to-event context)
>
> -	SurvITE (CFR-2): architecture with multiple IPM-terms that optimize for the balance of treatment groups at each time step (not wrt baseline) (slightly less naïve extension of [9] to time-to-event context).
>
> We would argue that, when including unrelated baseline algorithms in a setting as complex as ours, it becomes difficult to understand the sources of gain: is a difference in performance due to the used model architecture or ML method, due to the difference in handling (a subset of) the shifts, or a combination of the two? Therefore, we limited our comparisons to  (i) a selection of off-the-shelf survival models that are widely used in the survival analysis literature (e.g., Cox and random survival forests) and (ii) the only deep learning model that considers treatment effects in the survival setting (CSA-INFO [27]). Intuitively speaking, the Cox and LR-sep baseline acted as a reference point, marking the difficulty arising due to nonlinearities and nonproportional hazards.
>
> Based on the reviewer’s suggestion, we repeated the experiments using DeepHit [5] (which we chose since it does not rely on the proportional hazard assumption, as opposed to DeepSurv) on the Synthetic and Twins experiments. Here, we included a part of the experiments on S3 ($\eta = 3, \mathcal{P} = \{9,10\}$) and S4 ($\eta = 3, \mathcal{P} = \{9,10\}$) from Figure 3 and Table 1 in the manuscript for illustration:
>
>
>
> Table A. Performance of estimating the survival function (with T=10)
>
> | Methods                 | S3. ($\zeta = 3, \mathcal{P} = \\{9,10\\}$) | S4. ($\zeta = 3, \mathcal{P} = \\{9,10\\}$) |
> |-------------------------|-------------------------------------------|-------------------------------------------|
> |     Cox                 |     0.127 $\pm$ 0.002                     |     0.127 $\pm$ 0.002                     |
> |     RSF                 |     0.074 $\pm$ 0.005                     |     0.079 $\pm$ 0.005                     |
> |     LR-sep              |     0.112 $\pm$ 0.002                     |     0.115 $\pm$ 0.003                     |
> |     DeepHit             |     0.095 $\pm$ 0.012                     |     0.087 $\pm$ 0.013                     |
> |     CSA                 |     0.155 $\pm$ 0.005                     |     0.147 $\pm$ 0.020                     |
> |     SurvITE (no IPM)    |     0.086 $\pm$ 0.008                     |     0.088 $\pm$ 0.011                     |
> |     SurvITE (CFR-1)     |     0.084 $\pm$ 0.003                     |     0.097 $\pm$ 0.009                     |
> |     SurvITE (CFR-2)     |     0.059 $\pm$ 0.003                     |     0.085 $\pm$ 0.017                     |
> |     SurvITE             |     0.055 $\pm$ 0.007                     |     0.063 $\pm$ 0.010                     |
>
> Table B. Performance of estimating the difference in survival functions (with T=10)
>
> |     Methods             |     S3. ($\zeta = 3, \mathcal{P} = \\{9,10\\}$    |     S4. ($\zeta = 3, \mathcal{P} = \\{9,10\\}$    |
> |-------------------------|---------------------------------------------------|---------------------------------------------------|
> |     Cox                 |     0.101$\pm$ 0.004                              |     0.099$\pm$ 0.004                              |
> |     RSF                 |     0.081$\pm$ 0.003                              |     0.084$\pm$ 0.004                              |
> |     LR-sep              |     0.096$\pm$ 0.003                              |     0.099$\pm$ 0.005                              |
> |     DeepHit             |     0.107$\pm$ 0.014                              |     0.095$\pm$ 0.007                              |
> |     CSA                 |     0.176$\pm$ 0.025                              |     0.148$\pm$ 0.011                              |
> |     SurvITE (no IPM)    |     0.071$\pm$ 0.011                              |     0.079$\pm$ 0.012                              |
> |     SurvITE (CFR-1)     |     0.068$\pm$ 0.009                              |     0.083$\pm$ 0.005                              |
> |     SurvITE (CFR-2)     |     0.061$\pm$ 0.009                              |     0.073$\pm$ 0.011                              |
> |     SurvITE             |     0.060$\pm$ 0.009                              |     0.063$\pm$ 0.004                              |
>
> Table C. Performance of estimating HTE-RMST in Table 1 of the manuscript
>
> | Methods                 | Twins (no censoring)     | Twins (no censoring)      | Twins (censoring)        | Twins (censoring)         |
> |-------------------------|--------------------------|---------------------------|--------------------------|---------------------------|
> |                         | L=30                     | L=180                     | L=30                     | L=180                     |
> |     Cox                 |     2.85 $\pm$ 0.10      |     20.33 $\pm$ 0.50      |     2.88 $\pm$ 0.09      |     20.60 $\pm$ 0.50      |
> |     RSF                 |     3.15 $\pm$ 0.07      |     22.42 $\pm$ 0.36      |     3.18 $\pm$ 0.08      |     22.62 $\pm$ 0.46      |
> |     LR-sep              |     2.94 $\pm$ 0.10      |     20.60 $\pm$ 0.53      |     2.94 $\pm$ 0.10      |     20.66 $\pm$ 0.52      |
> |     DeepHit             |     2.95 $\pm$ 0.28      |     20.89 $\pm$ 1.91      |     2.86 $\pm$ 0.09      |     20.69 $\pm$ 0.52      |
> |     CSA                 |     3.42 $\pm$ 0.12      |     26.20 $\pm$ 1.21      |     4.41 $\pm$ 0.54      |     47.79 $\pm$ 1.55      |
> |     SurvITE (no IPM)    |     2.80 $\pm$   0.10    |     19.80 $\pm$   1.01    |     2.85 $\pm$   0.22    |     20.00 $\pm$   1.07    |
> |     SurvITE (CFR-1)     |     2.68 $\pm$ 0.06      |     19.16 $\pm$ 0.37      |     2.67 $\pm$ 0.15      |     19.10 $\pm$ 0.85      |
> |     SurvITE (CFR-2)     |     2.61 $\pm$ 0.12      |     18.69 $\pm$ 0.64      |     2.69 $\pm$ 0.22      |     19.20 $\pm$ 1.44      |
> |     SurvITE             |     2.53 $\pm$ 0.09      |     18.34 $\pm$ 0.70      |     2.63 $\pm$ 0.10      |     18.76 $\pm$ 0.56      |
>
> Overall, we observe that DeepHit performs worse than the SurvITE architecture without IPM term, indicating that our model architecture alone is better suited for estimation of treatment-specific survival functions (note that [5] focused mainly on discriminative (predictive) performance, and not on the estimation of the survival function itself). Therefore, upon addition of the IPM-terms, the performance gap between SurvITE and DeepHit only becomes larger.

---

> > ### Comment · Reviewer_4Jiu · 2021-08-30
> > **Thanks**
> >
> > Thank you for the very detailed answer.
> >
> > I do not have further questions and I think the authors have managed to answer my concerns. The second part of the answer is very interesting and could be added to the problem, even if only in the supplementary for the readers.

---

> > > ### Author Response · Authors · 2021-08-30
> > > **Thanks**
> > >
> > > Dear Reviewer  4Jiu,
> > >
> > > We are sincerely grateful for your time and energy in the review process. In light of your satisfaction with our response, we wonder whether the reviewer would kindly consider revising the rating.
> > >
> > > Thank you,
> > > Paper 4461 Authors

---

> ### Author Response · Authors · 2021-08-18
> **Dear Reviewer 4Jiu**
>
> Once again, we would like to thank you for your invaluable feedback! We were wondering whether our response from August 10 has sufficiently addressed your concerns. If you have any leftover comments or concerns, please let us know - we would be happy to do our utmost to address them!

---

> ### Author Response · Authors · 2021-08-23
> **Did our response address your concerns?**
>
> Dear Reviewer 4Jiu,
>
> We are sincerely grateful for your time and energy in the review process!
>
> Given the limited time left in this response phase, we wanted to make a final enquiry whether our response has addressed your concerns. We would also be eager to follow up on any additional comments you may have!
>
> Thank you!
>
> Paper4461 Authors

---

### Official Review · Reviewer_ukxC · 2021-07-21

**Rating:** 6
**Confidence:** 4

**Summary:**

In this paper, the authors present a framework for heterogeneous treatment effect estimation for time-to-event outcomes. In recent times, both areas have been explored independently and this paper proposes an approach for the censored time-to-event setting. SurvITE (individual treatment effect) and SurvHTE (heterogeneous treatment effect). Authors present experimental results for the same comparing against ablations and survival analysis baselines.

**Limitations And Societal Impact:**

Yes

**Main Review:**

**originality** - the proposed problem setup brings together significant body of work in both areas (survival analysis and heterogeneous/individual treatment effect estimation) together. The proposed formulation in Eq (6) uses ideas similar to those proposed in 8,9,10 and 32 from the references.

**quality and clarity** I do believe the draft appears to be technically sound and well presented in general. My only concern is with the way the IPM is applied in this paper. While authors do claim that this help obtaining shift-invariant representations, it is not directly clear for me on how penalizing IPM for each (a,$\tau$) combination w.r.t the baseline is more better than doing it across the treatments explicitly. It might help to improve the writeup here, as I do think this is critical for the paper.

In addition, I do think the experimental analysis section does appear incomplete.  I also don't completely agree with authors claim on line 44 that reference 27 is doing plain regression on targets. I think a more comprehensive discussion is needed w.r.t this paper in the experiments also if possible.

**significance** - Making a contribution in this space of HTE for time-to-event survival outcomes is very much needed.

**Post author feedback**

I have gone through the author response and I have changed my score for this paper. I do believe clarifying these concerns will improve the paper quality.

**Time Spent Reviewing:**

2.5 hours

---

> ### Author Response · Authors · 2021-08-10
> **Responses to the comments on IPM using the baseline population and comparison to regression-based survival models**
>
> Thank you for your thoughtful comments and suggestions! Below, we discuss responses to your main comments.
>
> [1. IPM using the Baseline Population]
>
> Allow us to reiterate that, as visualized in the DAG in Figure 1, there are three distinct sources of covariate shift when estimating treatment-specific hazard functions, which make the problem of obtaining reliable treatment effect estimates in our context unique and challenging. We tackle the problem of adjusting for these potential covariate shifts by finding representations that minimize the IPM term between the distribution at each $(a, \tau)$ combination and the (marginal) distribution at baseline. This will automatically adjust for all three sources of shift.
>
> When applying the IPM-term to encourage only the distributions $(0, \tau)$ and $(1, \tau)$ at each time-step $\tau$ to be similar, this corrects only for shifts due to (i) confounding at baseline, (ii) treatment-induced differences in censoring and (iii) treatment-induced differences in events. It will, however, not allow handling the event- and censoring-induced shifts that occur regardless of treatment status. As an illustrative example, assume an RCT setting (i.e., no confounding at baseline) in which older patients are more likely to be censored, equally across both treatment arms. In this example, there will be no difference in patient populations across the treatment arms over time, but the population left for analysis will get younger at each time-step $\tau$ (i.e., there is censoring-induced covariate shift over time). Thus, balancing only the treatment arms at each time-step cannot correct for all sources of covariate shift over time. To correct this censoring-induced shift, balancing towards the baseline population is necessary. The same argument holds for event-induced shift.  We would like to thank you for pointing out that this important point needs more emphasis, and will make this more clear in the updated manuscript.
>
> Further, we note that we actually verified this intuition empirically: in our experiments, we indeed consider two variants of SurvITE that do balancing only across treatment arms, and not wrt the baseline population (denoted as SurvITE (CFR-1) and SurvITE (CFR-2) in the manuscript, line 339-342). We thereby confirmed that naive balancing across treatment arms is not as effective as using the baseline population as a target, especially at the later time steps where the effects of bias due to changes over time worsens.
>
>
> [2. Comparison to Regression-Based Survival Models]
>
> We would like to begin by pointing out that the primary goal of survival analysis is to estimate survival functions, i.e. $S(t|x) = P(T \geq t|x)$, which is the probability of an event of interest occurring at or after a given time $t$. Survival models (e.g., our proposed method) that learn hazard functions from time-to-event data are able to directly compute estimated survival functions through the definition in equation (2) in the manuscript.
>
> In our paper, we used the term ‘regression-based survival models’ to describe models that do not estimate hazard or survival functions, but instead learn to model the event time $T$, e.g., by modeling the expected outcome $E[T|X=x]$, or by learning to generate samples $T \sim \mathbb{P}(\cdot|X=x)$ (as CSA-INFO [27] does). We therefore did not intend to imply that CSA-INFO performs ‘plain regression on targets’ (indeed, CSA-INFO is much more sophisticated than that, as it has mechanisms to handle both censoring and selection bias), but rather wanted to emphasize that CSA-INFO does not model survival- or hazard function directly. We will make this distinction more clear in the updated manuscript.
>
> Finally, to transform the outputs of regression-based models to estimates of survival functions, one needs to either (i) make explicit modeling assumptions on the time-to-event process, to link $E[Y|X=x]$ to $S(t|x)$ or (ii) create a Monte-Carlo approximation of $S(t|x)$ by sampling $T \sim \mathbb{P}(\cdot|X=x)$ from the learned generator. This is why we had to perform Monte-Carlo sampling to approximate survival functions for CSA-INFO in our experiments (see also line 336-337). We will clarify this further in the updated manuscript.

---

> ### Author Response · Authors · 2021-08-18
> **Dear Reviewer ukxC**
>
> Once again, we would like to thank you for your invaluable feedback! We were wondering whether our response from August 10 has sufficiently addressed your concerns. If you have any leftover comments or concerns, please let us know - we would be happy to do our utmost to address them!

---

> ### Author Response · Authors · 2021-08-23
> **Did our response address your concerns?**
>
> Dear Reviewer ukxC,
>
> We are sincerely grateful for your time and energy in the review process!
>
> Given the limited time left in this response phase, we wanted to make a final enquiry whether our response has addressed your concerns. We would also be eager to follow up on any additional comments you may have.
>
> Thank you!
>
> Paper4461 Authors

---

### Decision · Program_Chairs · 2021-09-27

**Decision:**

Accept (Poster)

**Comment:**

The authors formalize heterogeneous treatment effect estimation from time-to-event data under empirical risk minimization. This is a natural extension of the prior work done for continuous and binary outcomes with the added challenges from to the time-to-event data such as censoring bias. The paper being a straightforward follow the recipe type of paper doesn't detract from its quality. The main thing I'd suggest would be complete the references in the related work for ML methods to include things like:

@inproceedings{avati2020countdown,
  title={Countdown regression: sharp and calibrated survival predictions},
  author={Avati, Anand and Duan, Tony and Zhou, Sharon and Jung, Kenneth and Shah, Nigam H and Ng, Andrew Y},
  booktitle={Uncertainty in Artificial Intelligence},
  pages={145--155},
  year={2020},
  organization={PMLR}
}

and

@inproceedings{ranganath2016deep,
  title={Deep survival analysis},
  author={Ranganath, Rajesh and Perotte, Adler and Elhadad, No{\'e}mie and Blei, David},
  booktitle={Machine Learning for Healthcare Conference},
  pages={101--114},
  year={2016},
  organization={PMLR}
}